# Deep Markov Factor Analysis:
# Towards Concurrent Temporal and Spatial Analysis of fMRI Data

**Amirreza Farnoosh**
Augmented Cognition Lab (ACLab)
Department of Electrical & Computer Engineering
Northeastern University
Boston, MA 02115
farnoosh.a@northeastern.edu

**Sarah Ostadabbas**
Augmented Cognition Lab (ACLab)
Department of Electrical & Computer Engineering
Northeastern University
Boston, MA 02115
ostadabbas@ece.neu.edu

## Abstract

Factor analysis methods have been widely used in neuroimaging to transfer high dimensional imaging data into low dimensional, ideally interpretable representations. However, most of these methods overlook the highly nonlinear and complex temporal dynamics of neural processes when factorizing their imaging data. In this paper, we present deep Markov factor analysis (DMFA), a generative model that employs Markov property in a chain of low dimensional temporal embeddings together with spatial inductive assumptions, all related through neural networks, to capture temporal dynamics in functional magnetic resonance imaging (fMRI) data, and tackle their high spatial dimensionality, respectively. Augmented with a discrete latent, DMFA is able to cluster fMRI data in its low dimensional temporal embedding with regard to subject and cognitive state variability, therefore, enables validation of a variety of fMRI-driven neuroscientific hypotheses. Experimental results on both synthetic and real fMRI data demonstrate the capacity of DMFA in revealing interpretable clusters and capturing nonlinear temporal dependencies in these high dimensional imaging data.

## 1 Introduction

Functional magnetic resonance imaging (fMRI) has been extensively used in cognitive neuroscience to study brain structures and their interactions at rest or during a cognitive task [Glover, 2011]. fMRI also provides insights into how brain's functional connectivity changes during different experimental conditions [Preti et al., 2017, Azari et al., 2020]. However, due to high dimensional nature of fMRI data (tens of thousands of voxels in few seconds-long sessions), analyzing functional connectivity of brain could get very challenging [Turk-Browne, 2013]. Most classical methods employ region of interest (ROI)-based approaches to reduce size of data for processing [Poldrack, 2007, Etzel et al., 2009, Farnoosh and Soltanian-Zadeh, 2017, Gadgil et al., 2020]. However, averaging across many voxels within each region could wash out signals from small number of task-relevant voxels with noise from non-activated voxels, therefore results in a loss of information [Poldrack, 2007].

35th Conference on Neural Information Processing Systems (NeurIPS 2021)

Recently, few approaches have been proposed, based on probabilistic generative models, for topographic factorization of fMRI data into a weighted summation of few localized activation sources (i.e., temporal weights and topographic spatial factors), among which topographic factor analysis (TFA) [Manning et al., 2014b] and its multi-subject extension, hierarchical TFA (HTFA) [Manning et al., 2018], and neural TFA (NTFA) [Sennesh et al., 2020] are the most noted ones. This factorization serves as a necessary preparation step for subsequent statistical analysis that can effectively characterize subject- and stimulus-level variations and reveal task- or cognitive state-related networks in brain. However, TFA approaches assume a prior in which temporal weights are conditionally independent as a function of time, which means they do not encode temporal dynamics. Given the non-linearity and complex time-dependencies inherent in fMRI, a model is required that can capture and represent these dependencies.

In this paper, we propose deep Markov factor analysis (DMFA)[1], a Bayesian model for factorization of fMRI data that learns a deep generative Markovian prior to reason about nonlinear temporal dynamics. This is realized by a chain of low dimensional temporal embeddings related through neural networks. This prior is further augmented by a discrete latent for multimodal dynamical estimation, and clustering subject- and task-level variations directly in its low dimensional temporal embedding. To accommodating high spatial dimensionality, DMFA generatively parameterize spatial factors from a low dimensional spatial latent through neural networks.

We evaluate the performance of DMFA on a synthetic and two real large-scale fMRI datasets. Our experiments demonstrate that DMFA uncovers meaningful clusters in these data and achieves better predictive performance for unseen data relative to the state-of-the-art.

## 2 Related Work

**Factor Analysis in fMRI:** Factor analysis in neuroimaging includes a wide range of approaches for reducing data dimensionality to facilitate their interpretability and computational tractability. Principal component analysis (PCA) [Pearson, 1901] and independent component analysis (ICA) [Comon et al., 1991] are among the most well-known classical factor analysis methods. To accommodate tensor data and mitigate scalability issues, multilinear versions of PCA and ICA have been proposed in Vasilescu and Terzopoulos [2005], Beckmann and Smith [2005], Cichocki [2011], Richard and Montanari [2014], Hopkins et al. [2015]. Specifically, for multi-subject fMRI study, Lee et al. [2008] proposed independent vector analysis (IVA) and Richard et al. [2020] developed MultiView ICA to model shared responses. Likewise, Chen et al. [2015] developed a shared response model (SRM) for aggregating multi-subject fMRI data and highlighting group differences, and Van Kesteren and Kievit [2021] incorporated structured residuals into the exploratory factor analysis (EFA) framework. Karahanoğlu and Van De Ville [2015] deconvolved hemodynamic response from rest fMRI time series and then performed temporal clustering on the resulting whole-brain innovation signals to recover the corresponding spatial patterns. However, spatial factors obtained by these methods are unstructured, often have the same size as the images in the original dataset, and may include many small and large voxel clusters across the brain, therefore are not directly interpretable [Manning et al., 2014b]. More important, these methods are permutation invariant along temporal dimension (i.e., do not assume any relationships between temporal dimensions), therefore do not model temporal dynamics [Yu et al., 2016].

To enhance spatial interpretability, Manning et al. [2014b] introduced topographic factor analysis (TFA), a probabilistic technique that casts each brain image as a weighted sum of *Gaussian* spatial factors. In this way, each Gaussian blob (a.k.a topographic factor) can be thought of as a brain region for which activation levels are estimated over time, and is easily interpreted through its set of parameters. Later on, authors proposed a hierarchical version of TFA (HTFA) [Manning et al., 2018] to incorporate data from multiple subjects and enable hypothesis testing across subjects by applying hierarchical Gaussian priors for global and subject-specific parameters of temporal and topographic factors. These linearly-dependent hierarchical Gaussian priors, though, favor estimation of unimodal distributions and limit the model's expressivity. Recently, Sennesh et al. [2020] proposed neural TFA (NTFA) for task fMRI, which extends TFA by incorporating neural networks onto its framework. NTFA assumes separate latent embeddings for participants and stimuli and map them into the temporal and spatial latents with neural networks. However, methods in this line of work

---

[1] The source code is available at https://github.com/ostadabbas/DMFA

essentially assume a prior in which temporal weights are conditionally independent as a function of time, which means that they do not encode temporal dynamics.

**Dynamical Factorization:** A number of matrix/tensor factorization approaches have been proposed for modeling temporal dynamics in sequential data. These methods employ linear-Gaussian state-space models [Sun et al., 2014], autoregressive temporal regularizer [Bahadori et al., 2014, Yu et al., 2016, Takeuchi et al., 2017], and multilinear dynamical systems [Rogers et al., 2013, Cai et al., 2015, Jing et al., 2018]. In contrast to these methods which provide point estimates, *Bayesian* dynamical factorization has been proposed in Xiong et al. [2010], Charlin et al. [2015], Sun and Chen [2019], in which linear temporal dynamics are applied to factor latents. While some of these methods have been successful in dynamical modeling of sequential data, they are less effective for high dimensional spatial data like fMRI as they do not explicitly adopt any structural constraints for spatial factors. Moreover, their linear dynamical assumptions lack the capacity to characterize nonlinear dependencies.

Motivated by recent advances in deep learning, several studies have implemented neural networks into Gaussian state space models for nonlinear dynamical modelling [Krishnan et al., 2015, Watter et al., 2015, Chung et al., 2015, Karl et al., 2017, Krishnan et al., 2017, Fraccaro et al., 2017, Becker et al., 2019, Farnoosh et al., 2021]. A common practice is to learn a temporal latent model followed by a mapping to data space, a.k.a. a decoder, and an encoder for *amortized* variational inference. However, this encoding/decoding framework is not tractable in very high dimensional spatial data like fMRI. To be more specific, it is computationally intensive to feed high dimensional data for amortized estimation, or map to them directly from a latent space (see Section 4.1 for a detailed discussion). Also, several studies have employed recurrent neural networks (RNNs) for temporal analysis of fMRI data [Hjelm et al., 2018, Dvornek et al., 2018, Yan et al., 2019, Wang et al., 2019]. RNN-based methods are not probabilistic and can only process reduced dimensional ROI data. Although convolutional RNN models can capture both temporal and spatial correlations, by construction they do not provide the familiar spatial correlation maps that we expect in neuroimaging analysis and that are necessary for neuroscientific interpretability. In addition, RNN model are harder to train on long sequences due to the vanishing/exploding gradient problem as they feed parameters sequentially. While this problem is much alleviated in LSTM, GRU, and Transformer models, these architectures still result in a huge computational graph and intense GPU memory consumption on long sequences, which leads to very long training times. In contrast, DMFA, as will be explained below, samples from a distinct posterior distribution at each time point and can fit arbitrarily long sequences without any issues (by allowing parallel computations over time), and at the same time infers smooth spatial factors automatically.

# 3 Deep Markov Factor Analysis (DMFA)

Let's assume a corpus of $N$ fMRI data $\{Y_n \mid n \in [1{:}N]\}$ for $Y_n \in \mathbb{R}^{T \times V}$, where $T$ and $V$ are the number of time points and voxels, respectively. DMFA defines a hierarchical dynamical deep generative model over this corpus in a factorization framework:

$$Y_n \sim \mathrm{Norm}(W_n^\top F_n, \, \sigma^{\mathrm{Y}} I),$$
$$W_n \sim \mathrm{Norm}(\mu_\theta^{\mathrm{W}}(Z_n^{\mathrm{W}}), \sigma_\theta^{\mathrm{W}}(Z_n^{\mathrm{W}})),$$
$$F_n = \left[ f_{k,v} := \mathrm{RBF}_k(v; \rho_k, \gamma_k) \right],$$
$$\rho, \gamma \sim \mathrm{Norm}(\mu_\theta^{\mathrm{F}}(z_n^{\mathrm{F}}), \sigma_\theta^{\mathrm{F}}(z_n^{\mathrm{F}})),$$
$$z_n^{\mathrm{F}} \sim \mathrm{Norm}(0, I),$$
$$Z_n^{\mathrm{W}} \sim p_\theta(Z_n^{\mathrm{W}} \mid \mathrm{C}_n),$$
$$\mathrm{C}_n \sim \mathrm{Cat}(\pi).$$

where $F_n \in \mathbb{R}^{K \times V}$ are $K \ll V$ spatial factors and $W_n \in \mathbb{R}^{K \times T}$ are their associated temporal weights, consistent with a factorization framework. $p_\theta(Z_n^{\mathrm{W}} \mid \mathrm{C}_n)$ is a deep generative Markovian prior over a set of low dimensional temporal latents $Z_n^{\mathrm{W}} = \{z_{n,t}^{\mathrm{W}} \mid t \in [0{:}T]\}$

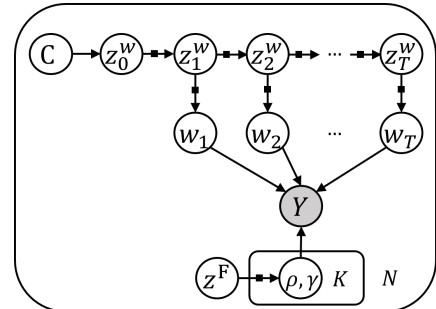

Figure 1: Graphical model representation for deep Markov factor analysis (DMFA). DMFA incorporates a deep generative Markovian prior $p_\theta(z_t^{\mathrm{W}} | z_{t-1}^{\mathrm{W}})$ to represent temporal variations in $W$, which is conditioned on a discrete latent, C, for data clustering. The $K$ spatial factors are conditioned on a shared latent, $z^{\mathrm{F}}$. Latent nodes and observations are represented by solid and gray-shaded circles, respectively. The solid black squares denote nonlinear mappings parameterized by neural networks.

conditioned on a discrete latent $C_n$ for clustering. The temporal weights $W_n$ are sampled from a Gaussian distribution whose mean and covariance are a function of $Z_n^W$ and are parameterized by neural networks $\mu_\theta^W$ and $\sigma_\theta^W$, respectively. The spatial factors $F_n$ are determined by a set of $K$ radial basis functions (RBFs) whose parameters $\{\rho, \gamma\}$ are sampled from a Gaussian distribution parameterized by neural networks $\mu_\theta^F$ and $\sigma_\theta^F$ from a low dimensional spatial latent $z_n^F$. All networks have parameters, which are collectively denoted by $\theta$. Finally, $\sigma^Y$ denotes observation noise. The graphical model for DMFA is depicted in Fig. 1.

**Clustering Latent:** We assume that each fMRI data $Y_n$ belongs to a specific cluster $s \in [1{:}S]$ which is declared by its associated discrete latent $C_n \sim \text{Cat}(\pi)$ sampled from a categorical prior with assignment probability vector $\pi$. This discrete latent conditions the first temporal latent $z_{n,0}^W$ resulting in a Gaussian mixture distribution over this temporal latent:

$$p_\theta(z_{n,0}^W | C_n = s) = \text{Norm}(\mu_s, \Sigma_s),$$

where $\mu_s$ and $\Sigma_s$ are determined by cluster assignment.

**Deep Generative Markovian prior:** We assume that the temporal latents $Z_n^W$ are related through a Markov chain, for which the transition probability $p_\theta(z_{n,t}^W | z_{n,t-1}^W) = \text{Norm}(\mu_\theta^Z(z_{n,t-1}^W), \sigma_\theta^Z(z_{n,t-1}^W))$ is a Gaussian distribution whose mean and covariance are parameterized by neural networks from $z_{n,t-1}^W$. We further blend the estimated mean from neural network with a linear transformation of $z_{n,t-1}^W$ to support both nonlinear and linear transitions (we dropped $n$ for brevity):

$$\mu_\theta^Z(z_{t-1}^W) = (1 - g) \odot \mu_\theta^{Z,L}(z_{t-1}^W) + g \odot \mu_\theta^{Z,NL}(z_{t-1}^W)$$

where $\mu_\theta^{Z,L}(\cdot)$ represents a linear mapping, $\mu_\theta^{Z,NL}(\cdot)$ is the nonlinear mapping of neural network, and $g \in [0, 1]$ is a weighting vector estimated from $z_{t-1}^W$ with another neural network.

**Temporal Weights & Spatial Factors:** As with the transition model, we assumed Gaussian distributions for temporal weights, such that their means and covariances are a function of their associated temporal latents:

$$w_{n,t} \sim \text{Norm}(\mu_\theta^W(z_{n,t}^W), \sigma_\theta^W(z_{n,t}^W)),$$

where $\mu_\theta^W$ and $\sigma_\theta^W$ are neural network functions. The high spatial dimensionality in fMRI data encourages the need for a hierarchical analysis that summarizes spatial factors with fewer parameters. Therefore, consistent with Manning et al. [2014b,a, 2018], we represent each spatial factor as a radial basis function (i.e., a Gaussian blob):

$$f_{kv} := \text{RBF}_k(v; \rho_k, \gamma_k) = \exp\left(-\frac{\|p_v - \rho_k\|^2}{\exp(\gamma_k)}\right),$$

where $p_v \in \mathbb{R}^3$ denotes the position of voxel with index $v$, and $\rho_k, \gamma_k \in \mathbb{R}^3$ denote the center and extent of the Gaussian blob, respectively. The set of all RBF parameters $\{\rho_k, \gamma_k \,|\, k \in [1{:}K]\} \sim \text{Norm}(\mu_\theta^F(z_n^F), \sigma_\theta^F(z_n^F))$ are sampled from a Gaussian distribution whose mean and covariance are a function of the shared spatial latent $z_n^F$. Introducing $z_n^F$ as a low dimensional spatial embedding encourages estimation of a multi-modal distribution among spatial factors.

## 3.1 Learning DMFA with Variational Inference

We train DMFA with stochastic variational methods [Hoffman et al., 2013, Ranganath et al., 2013]. These methods approximate the posterior $p_\theta(W, Z^W, z^F, C, \rho, \gamma | Y)$ using a variational distribution $q_\phi(W, Z^W, z^F, C, \rho, \gamma)$, where $\phi$ denotes parameters of the variational model, by maximizing a lower bound on the log-likelihood of the data:

$$\mathcal{L}(\theta, \phi) = \log p_\theta(Y) - \text{KL}\big(q_\phi(W, Z^W, z^F, C, \rho, \gamma) \,\|\, p_\theta(W, Z^W, z^F, C, \rho, \gamma | Y)\big).$$

By maximizing this bound with respect to the parameters $\theta$ and $\phi$, we learn a deep generative model and perform Bayesian inference, respectively.

**Parameterizing Variational Distribution:** We assume a fully factorized variational distribution and introduce trainable variational parameters for each data as mean and covariance of a diagonal Gaussian:

$$q_\phi(W, Z^W, z^F, C, \rho, \gamma) = \prod_{n=1}^{N} q(C_n) q(z_{n,0}^W) q(z_n^F) \prod_{k=1}^{K} q(\rho_{n,k}, \gamma_{n,k}) \prod_{t=1}^{T} q(z_{n,t}^W) q(w_{n,t}),$$

Table 1: Network architectures in DMFA. The fully connected (FC) layers parameterize the three Gaussian distributions in the generative model: $p_\theta(z_t^{\text{W}}|z_{t-1}^{\text{W}})$, $p_\theta(w_t|z_t^{\text{W}})$ and $p_\theta(\rho,\gamma|z^{\text{F}})$. The second row denotes the inputs to neural networks.

| Layer \ Model | $\boldsymbol{\mu}_{\boldsymbol{\theta}}^{\text{Z}}, \boldsymbol{\sigma}_{\boldsymbol{\theta}}^{\text{Z}} : \mathbb{R}^2 \to \mathbb{R}^2$ | $\boldsymbol{\mu}_{\boldsymbol{\theta}}^{\text{W}}, \boldsymbol{\sigma}_{\boldsymbol{\theta}}^{\text{W}} : \mathbb{R}^2 \to \mathbb{R}^K$ | $\boldsymbol{\mu}_{\boldsymbol{\theta}}^{\text{F}}, \boldsymbol{\sigma}_{\boldsymbol{\theta}}^{\text{F}} : \mathbb{R}^2 \to \mathbb{R}^{6K}$ |
|---|---|---|---|
| Input | $z_{t-1}^{\text{W}} \in \mathbb{R}^2$ | $z_t^{\text{W}} \in \mathbb{R}^2$ | $z^{\text{F}} \in \mathbb{R}^2$ |
| 1 | FC $2 \times D_t$ PReLU | FC $2 \times D_e$ PReLU | FC $2 \times 4$ PReLU |
| 2 | FC $D_t \times 2$ Sigmoid | FC $D_e \times 2D_e$ PReLU | FC $4 \times 8$ PReLU |
| 3 | $g \in [0,1]^2$ | FC $2D_e \times 2K$ | FC $8 \times 6K$ |
| 4 | FC $2 \times D_t$ PReLU | $\mu^{\text{W}}, \log\sigma^{\text{W}} \in \mathbb{R}^K$ | $\mu^{\text{F}} \in \mathbb{R}^{6K}$ |
| 5 | FC $D_t \times 2$ | | FC $2 \times 4$ PReLU |
| 6 | $\mu^{\text{Z,NL}} \in \mathbb{R}^2$ | | FC $4 \times 8$ PReLU |
| 7 | PReLU FC $2 \times 2$ | | FC $8 \times 6K$ |
| 8 | $\log\sigma^{\text{Z}} \in \mathbb{R}^2$ | | $\log\sigma^{\text{F}} \in \mathbb{R}^{6K}$ |
| 9 | FC $2 \times 2$ | | |
| 10 | $\mu^{\text{Z,L}} \in \mathbb{R}^2$ | | |

where $q(\cdot)$ for continuous latents are parametric Gaussian distributions. Note that due to the high dimensionality of fMRI data ($V \gg N$, i.e., number of voxels are much larger than number of data), we do not utilize an amortized inference (where a single parametric function maps each data to a set of variational parameters) as it could be computationally very expensive and result in over-fitting (see Section 4.1). Although we can define variational parameters for categorical distributions $q(\text{C}_n)$, we approximate it with posterior $p(\text{C}_n|z_{n,0}^{\text{W}})$ to compensate information loss induced by the mean-field approximation:

$$q(\text{C}_n = s) \simeq p(\text{C}_n = s|z_{n,0}^{\text{W}}) = \frac{p(\text{C}_n = s)p(z_{n,0}^{\text{W}}|\text{C}_n = s)}{\sum_{s=1}^{S} p(\text{C}_n = s)p(z_{n,0}^{\text{W}}|\text{C}_n = s)}$$

This approximation has a two-fold advantage: spares the model additional trainable parameters for variational distribution, and further links variational parameters of $q_\phi(z_{n,0}^{\text{W}})$ to generative parameters of $p_\theta(z_{n,0}^{\text{W}})$ and $p_\theta(c)$, hence results in a more robust learning and inference algorithm.

**Derivation of Evidence Lower Bound (ELBO):** Once the generative model and variational distribution are determined, we can derive the ELBO which contains a reconstruction term and regularization terms for each latent to bring their variational posterior as close as possible to their generative prior:

$$\mathcal{L}_n(\theta,\phi) = \mathbb{E}_{q_\phi(W_n)q_\phi(\rho_n,\gamma_n)}\Big[\log p_\theta(Y_n|W_n,F_n)\Big] \qquad \text{(Likelihood)}$$

$$- \sum_{\text{C}_n} q_\phi(\text{C}_n)\,\text{KL}\big(q_\phi(z_{n,0}^{\text{W}})||p_\theta(z_{n,0}^{\text{W}}|\text{C}_n)\big) - \text{KL}\big(q_\phi(\text{C}_n)||p(\text{C})\big) \qquad \text{(Clusters)}$$

$$- \sum_t \mathbb{E}_{q_\phi(z_{n,t-1}^{\text{W}})}\Big[\text{KL}\big(q_\phi(z_{n,t}^{\text{W}})||p_\theta(z_{n,t}^{\text{W}}|z_{n,t-1}^{\text{W}})\big)\Big] \qquad \text{(Markov Transitions)}$$

$$- \sum_t \mathbb{E}_{q_\phi(z_{n,t}^{\text{W}})}\Big[\text{KL}\big(q_\phi(w_{n,t})||p_\theta(w_{n,t}|z_{n,t}^{\text{W}})\big)\Big] \qquad \text{(Weights)}$$

$$- \sum_k \mathbb{E}_{q_\phi(z_n^{\text{F}})}\Big[\text{KL}\big(q_\phi(\mu_{n,k},\gamma_{n,k})||p_\theta(\mu_{n,k},\gamma_{n,k}|z_n^{\text{F}})\big)\Big] - \text{KL}\big(q_\phi(z_n^{\text{F}})||p(z^{\text{F}})\big). \qquad \text{(Factors)}$$

We compute Monte Carlo estimate of gradient of the ELBO using a reparameterized sample from the variational distribution of continuous latents. For the discrete latent, $\text{C}_n$, we compute the expectations over $q_\phi(\text{C}_n)$ by summing over all the possibilities, hence no sampling is performed. We analytically calculate the KL terms of ELBO for both multivariate Gaussian and categorical distributions, which leads to lower variance gradient estimates and faster training as compared to e.g., noisy Monte Carlo estimates often used in the literature.

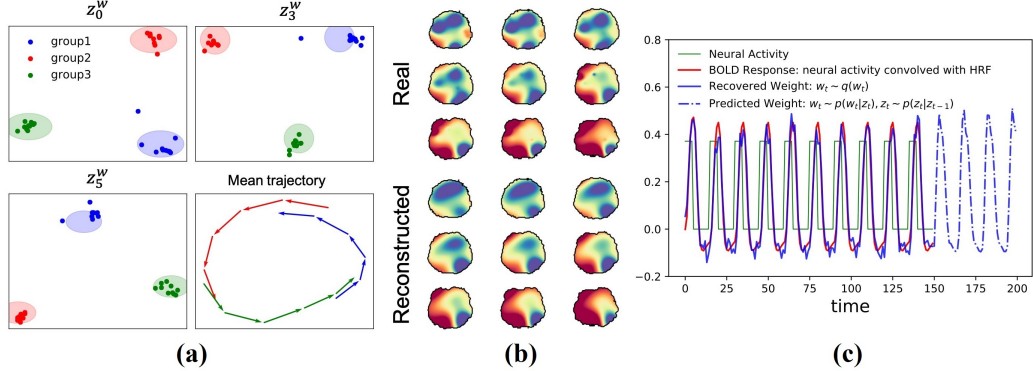

Figure 2: **(a)** DMFA recovered the three clusters of activation in our synthetic fMRI dataset, unsupervised. The mean dynamical trajectory (composed of three consecutive rotational dynamics) shows the inferred trajectory of each cluster mean over time in the temporal latent and is consistent with the periodic activation of sources in data clusters. **(b)** Real and reconstructed brain images. **(c)** The learned generative model's predictions for a selected activation source show that DMFA encoded the nonlinear hemodynamic response function in its deep temporal generative model.

## 4    Training Details

We described the network architectures of DMFA in Table 1, where $D_t$ and $D_e$ are the dimensions of hidden layers for $(\mu_\theta^z, \sigma_\theta^z)$ and $(\mu_\theta^w, \sigma_\theta^w)$, respectively. We did all the programming in PyTorch v1.3 and used Adam optimizer with learning rate of $0.01$. We initialized all the parameters randomly except for locations of Gaussian blobs for which we set initial values to local extrema in their averaged fMRI data. We clipped Gaussian blob parameters to the confines of brain if needed. We used a linear KL annealing schedule, [Bowman et al., 2016], which increased from $0.01$ to $1$ over $100$ epochs. We learned/tested all models on an Intel Core i7 CPU @3.7 GHz with 8 Gigabytes of RAM, which proves tractability of the learning process. Per-epoch training time varied from $0.1$ minutes in small datasets to $6.0$ minutes in larger experiments and $200$ epochs sufficed for most experiments.

### 4.1    Discussion of Parameter Count for DMFA

The number of learnable parameters for variational distribution in DMFA is $O(\mathrm{NTK})$. DMFA has $O(\mathrm{KD}_e + \mathrm{D}_t)$ parameters for its temporal generative model and $O(\mathrm{K})$ parameters for its spatial generative model. Note that the clustering latent, C, does not impose additional parameters to the variational distribution, while only adds $O(\mathrm{S})$ parameters to the temporal generative model.

While DMFA introduces extra features and more complex modeling assumptions for fMRI experiments compared to TFA methods of Manning et al. [2014b,a, 2018], (i.e., infers nonlinear temporal dynamics and performs clustering), we emphasize that it has the same order of parameters as these methods. TFA methods similarly have $O(\mathrm{NTK})$ parameters as they employ a fully factorized variational distribution. We also want to highlight that DMFA is tractable in very high dimensional spatial data like fMRI in contrast to nonlinear state-space models of Watter et al. [2015], Krishnan et al. [2017], Fraccaro et al. [2017], Karl et al. [2017], Becker et al. [2019]. The encoder/decoder structure in these works, i.e., $q_\phi(W_n|Y_n)/p_\theta(Y_n|W_n)$, immediately scales both generative and variational parameters to at least $O(\mathrm{KV})$, where $\mathrm{V} \sim 10^5 \gg \mathrm{NT}$ in fMRI data, hence causes extensive computational burden and more importantly overfitting. Furthermore, these methods do not learn a generative model for spatial factors, i.e., $p_\theta(\rho, \gamma)$, and as a result are not able to reason about subject-level variabilities in this respect. We overcome these challenges in DMFA by carefully designing our non-amortized variational inference and imposing functional form assumptions on spatial factors in a factorization framework. The proposed learning and inference algorithm keep generative parameters in $O\left(\mathrm{KD}_e + \mathrm{D}_t\right) \ll O(\mathrm{KV})$, and variational parameters in $O(\mathrm{NTK})$, where $\mathrm{NT} \sim 10^2 - 10^5$, yield an observation to parameter ratio of $O(\frac{\mathrm{NTV}}{\mathrm{NTK}}) = O(\frac{V}{K})$ for all the experiments, therefore permit an efficient learning process.

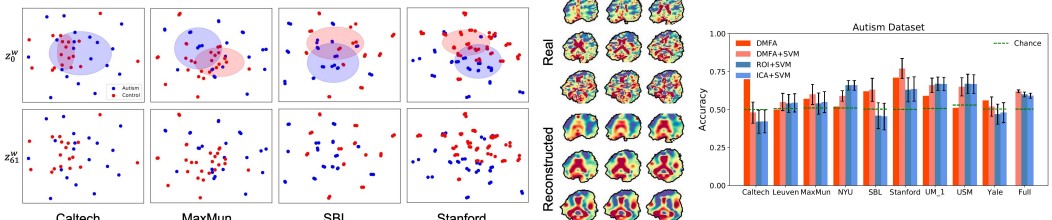

Figure 3: **(Left)** DMFA's clustering results show that the ASD and control groups can be partially separated. **(Middle)** Real and reconstructed brain images, showing the smoothing given by sparse factorization. **(Right)** A downstream classification task showed that `DMFA` and `DMFA+SVM` outperformed regions of interest (`ROI`)+`SVM` and `ICA+SVM` in the Caltech, SBL, Stanford, and Yale subsets of the dataset at $p$-value $< 0.005$, corrected for multiple comparisons. `ROI+SVM` and `ICA+SVM` performed better in the NYU subset.

## 5    Experimental Results

We analysed the performance of DMFA on a synthetic and two large-scale fMRI datasets. In the first experiment, we verified DMFA's capability in recovering the true clusters and capturing the underlying temporal dynamic in a synthesized fMRI dataset (visualized in Fig. 2). Next, we evaluated the performance of DMFA on a large scale resting-state fMRI data, Autism dataset [Craddock et al., 2013], and a task fMRI data, Depression dataset [Lepping et al., 2016]. We assessed the *clustering* feature of DMFA on both datasets in terms of disease and cognitive state separation tasks, visualized in Fig. 3 and Fig. 4. We further provided a quantitative comparison with two state-of-the-art Bayesian generative models for fMRI data, HTFA [Manning et al., 2018] and NTFA [Sennesh et al., 2020], and a deep state-space model, RKN [Becker et al., 2019], in terms of synthesis quality of the generative models on both datasets by computing held-out log-likelihood and prediction accuracy, respectively, in Table 2.

### 5.1    Synthetic fMRI

We generated synthetic fMRI data using a MATLAB package provided by Manning et al. [2014b], which is known to be useful for analysing fMRI models. The synthesized brain image for each trial (time point) is a weighted summation of a number of radial basis functions (spatial factors) randomly located in brain. This synthesized fMRI data is then convolved with a hemodynamic response function (HRF) and zero-mean Gaussian noise with a medium-level signal-to-noise ratio is added. Here, we considered 30 activation sources (spatial factors) randomly located in a standard `MNI-152-3mm` brain template with roughly $V = 270,000$ voxels and 150 trials. We randomly split these 30 activation sources into 3 groups, each having 10 of the Gaussian blobs. These three groups of sources are periodically activated according to some random weights, one after another, for 5 trials. We generated non-overlapping sequences of $T = 5$ time points from this synthetic fMRI data. This resulted in 10 data points for each activation group (i.e., $N = 30$). To train DMFA, we set $T = 5$, $K = 30$, $D_t = 2$, $D_e = 8$, $S = 3$, and $\sigma^Y = 0.01$. As depicted in Fig. 2 (a), DMFA was able to successfully recover the 3 clusters of activation that were present in this dataset totally unsupervised. The dynamical trajectory of each cluster mean in the temporal latent (i.e., $\mu_{z_t^w}|\mu_{z_{t-1}^w}, C$) is predicted sequentially from the learned generative model and is visualized in the bottom-right of Fig. 2 (a), and appears to be partitioned into three consecutive rotational dynamics. This is consistent with the periodic activations of sources in data clusters (which come in tandem). Predictions of the learned generative model for a selected activation source are visualized in Fig. 2 (d) for the next 50 time points, estimated as follows: $w_t \sim p(w_t|z_t)$, where $z_t \sim p(z_t|z_{t-1})$ for $t = \{151, \ldots, 200\}$. These predicted samples perfectly follow hemodynamic response function, confirming DMFA's capacity in capturing the underlying nonlinear HRF by using neural networks in its deep temporal generative model.

### 5.2    Autism Dataset

We used the publicly available preprocessed resting state fMRI (rs-fMRI) data from the Autism Brain Imaging Data Exchange (ABIDE) collected at 16 international imaging sites [Craddock et al., 2013]. This dataset includes rs-fMRI imaging from 408 individuals suffering from Autism Spectrum

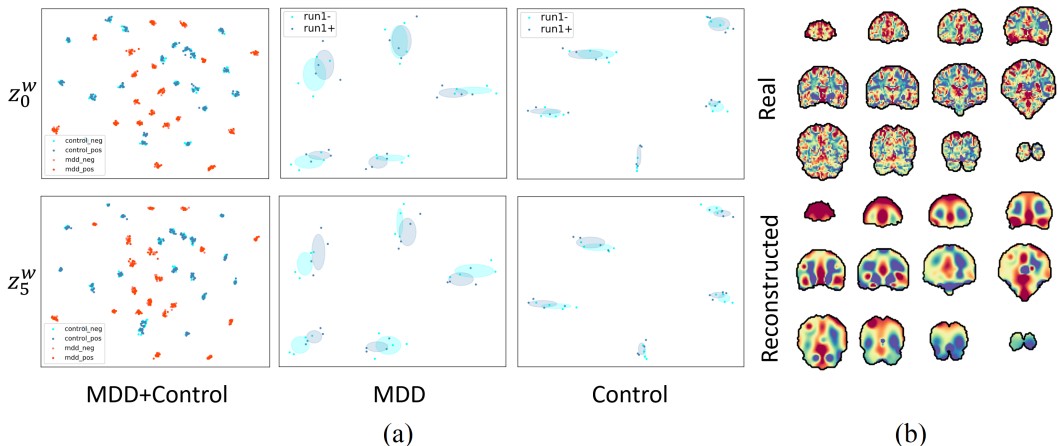

Figure 4: **(a), Left**: Training DMFA clustered together temporal latent variables associated with each subject without supervision, while partially separating clusters of points associated with the MDD group from those associated with the control group. The MDD group appears more concentrated into the center of the temporal latent space, while the control group have their temporal latent variables dispersed more broadly across the latent space. **(a), Middle, Right**: DMFA enabled us to partially separate "positive" and "negative" stimuli per-subject with Gaussian clusters. **(b)** Real and reconstructed brain images.

Disorder (ASD), and 476 typical controls. Each scan has $T = 145 \sim 315$ time points at TR = 2, and $V = 271,633$ voxels. We split the signals into sequences of 75 time points. We take two approaches to evaluate the performance of DMFA in separating ASD from control: (i) Cluster data directly in the low dimensional latent, $Z^{\mathrm{w}}$, using the clustering feature of DMFA (called `DMFA`), (ii) Extract functional connectivity matrices, [Hull et al., 2017], from learned temporal weights, $W$, followed by a 10-fold SVM for classification (called `DMFA+SVM`). As baselines, we performed a 10-fold SVM classification on extracted connectivity matrices from (i) averaged signals of 116 ROIs in automatic anatomical labeling (AAL) atlas [Kazeminejad and Sotero, 2019] (called `ROI+SVM`) and (ii) a hundred time courses obtained with nonlinear spatial ICA of Hyvarinen and Morioka [2017] (called `ICA+SVM`). Several studies have been done on this dataset to differentiate ASD group from control [Abraham et al., 2017, Parisot et al., 2017, Singh et al., 2017, Kazeminejad and Sotero, 2019, Wang et al., 2021, Sun et al., 2021], all of which using supervised methods, and could achieve accuracies up to 72% using the signals extracted from anatomically labeled regions in the brain by carefully splitting data to be as homogeneous as possible and reducing site-related variability.

We set $T = 75$, $K = 100$, $D_e = 15$, $D_t = 5$, $S = 2$, $\sigma^{\mathrm{Y}} = 0.01$, and trained DMFA for 200 epochs on the entire dataset (Full), and also datasets from 9 imaging sites (with more balanced datasets) separately: Caltech, Leuven, MaxMun, NYU, SBL, Stanford, UM, USM, Yale. As shown in Fig. 3 (Right), `DMFA` and `DMFA+SVM` outperformed baselines in Caltech, SBL, Stanford, and Yale (at $p$-value $< 0.005$, corrected for multiple comparisons), while `ROI+SVM` and `ICA+SVM` only performed better in NYU dataset. `DMFA+SVM` performed slightly better than baselines on the entire dataset (Please note that `DMFA` is a clustering approach, hence, no error bars are provided for it in Fig. 3). Clustering results for Caltech, Maxmun, SBL, and Stanford are shown in Fig. 3 (Left) in which ASD and control seems to be partially separable (see Fig. S1 in supplementary for more visualization results).

### 5.3 Depression Dataset

In this dataset [Lepping et al., 2016], 19 individuals with major depressive disorder (MDD) and 20 never-depressed control participants listened to standardized positive and negative emotional musical and nonmusical stimuli during fMRI scanning. Each participant underwent 3 musical, and 2 nonmusical runs each for 105 time points at TR=3 with $V = 353,600$ voxels. During each run, each stimulus type (positive, and negative) was presented for 33 seconds ($\sim 11$ time points) interleaved with instances of neutral tone of the same length. We discarded instances of neutral tone, and split each run into non-overlapping sequences of $T = 6$ time points in agreement with stimuli design (each stimuli block is split into two sequences). In other words, each run has 4 sequences associated with "positive stimuli", and 4 with "negative stimuli" resulting in a total of 8 data points for each run.

Table 2: Comparison of held-out log-likelihood and prediction accuracy. DMFA results in models with higher held-out likelihood and better prediction accuracy, therefore, is a better fit compared to the competitors.

| Model
Dataset | Negative Held-out Log-Likelihood (Nats) | | | | | Prediction (NRMSE%) | |
| | DMFA | NTFA | HTFA | Ablation | | DMFA | RKN |
| | | | | D̶MFA | D̶M̶F̶A̶ | | |
| --- | --- | --- | --- | --- | --- | --- | --- |
| Autism (Caltech) | **2.52** | 2.61 | 2.82 | 2.75 | 2.63 | **5.44** | 6.19 |
| Depression | **5.82** | 5.97 | 6.64 | 6.26 | 6.02 | **6.35** | 7.49 |
| Synthetic | **3.47** | 3.64 | 3.71 | 3.83 | 3.61 | **2.07** | 2.84 |

The best results are highlighted in bold fonts.

Likelihood values are scaled by $10^{-6}$, $10^{-5}$, and $10^{-5}$ for Autism, Depression, and Synthetic datasets, respectively.

In the *first experiment*, we trained DMFA on the entire musical runs ($N = 39 \times 3 \times 8 = 936$) by setting $T = 6$, $K = 100$, $D_e = 15$, $D_t = 5$, $\sigma^Y = 0.001$ for 200 epochs. The results are shown in Fig. 4 (a, Left). We observed that DMFA fully separated the data points associated with each subject into distinct clusters across the low dimensional temporal latent space. In other words, DMFA was able to re-unite pieces of signals associated with each subject without any supervision. More importantly, DMFA was able to partially separate the data points associated with MDD group from control. As seen in Fig. 4 (a, Left), the MDD group data points are fairly populated in the center of temporal latent while control group are dispersed across the latent space. However, DMFA was not able to meaningfully separate "negative" and "positive" music pieces in its low-dimensional latent space from a subject-level perspective, since the variation between runs of a subject dominates the stimulus-level variation. For this reason, in a *second experiment*, we focused on 5 subjects, and their first musical run from both MDD and control group and trained DMFA respectively. Again, as expected, data points from each subject were distinctly clustered in the latent space (see middle and right columns in Fig. 4 (a)). Additionally, DMFA was able to fit two partially separating Gaussians to "positive", and "negative" stimuli per subject. However, since the number of data points for each subject and run is limited, the significance of these clusters are not conclusive. A dataset with longer runs could possibly answer that.

## 5.4 Comparison with State-of-the-Art

We further evaluated DMFA against HTFA [Manning et al., 2018], an established probabilistic generative model for multi-subject fMRI analysis, which uses unimodal Gaussian priors for both temporal weights and spatial factor parameters, and its neural network-based extension NTFA [Sennesh et al., 2020] in terms of held-out log-likelihood, and a state-of-the-art deep state-space model, recurrent Kalman networks (RKN) [Becker et al., 2019], in terms of next-time-point prediction accuracy of test set (see Table 2). Held-out likelihood is an established metric for evaluating generative models (in lack of a ground-truth) and measures how probable an unseen data is under the generative model. For next-time-point prediction, we adopt a rolling prediction scheme as in Chen et al. [2019], and predict next time point on the test set sequentially from historical data using the generative model and spatial factors learned on the train set.

We split our data to train and test as follows. We used the Caltech site subset from autism dataset and split each subject's fMRI time series into two half (each with $T = 70$). We trained the models on the first half, and tested on the second half. For the depression dataset, we considered 4 sequences from each subject's run for training and tested on the remaining 4 sequences. We set $K = 100$ spatial factors for all the models and trained each with their default hyperparameters. For the RKN model, we fit AAL atlas ROI signals as it is intractable to feed raw high-dimensional data. We reported average normalized prediction error (NRMSE%) for both RKN and DMFA on AAL atlas regions to make a fair comparison. For the synthetic fMRI, we set $K = 30$ and picked half of the data for test, and used the 30 Gaussian blob activations as ROI signals for reporting prediction accuracy in RKN and DMFA. The results are shown in Table 2, which proves that DMFA resulted in models with higher likelihood on the test sets, hence it is a better fit when compared to NTFA and HTFA models. DMFA also resulted in better predictions on the test sets of autism, depression, and synthetic datasets when compared with RKN.

## 5.5 Ablation Study

We performed two ablation studies to further evaluate (and quantify) the impact of deep neural networks and Markov temporal apriori. First, we trained a version of DMFA in which we removed nonlinear activation functions (i.e., PReLU), denoted as ~~D~~MFA. As reported in Table 2, in our datasets, this resulted in a decrease in held-out likelihood. Also, the inferred temporal latents lacked the interpretable patterns seen in Fig. 3 and Fig. 4 in terms of clinical/cognitive separability as latents collapsed into the conditional Gaussian priors. Second, we trained a version of DMFA in which we removed the temporal connections (i.e., temporal transition model), denoted as D~~M~~FA. Again, this resulted in a drop in held-out likelihood as reported in Table 2, however, the temporal latents preserved their interpretable patterns.

## 6 Conclusion

We presented deep Markov factor analysis, DMFA, a new probabilistic model for robust factor analysis of high dimensional fMRI data. We employed a chain of low dimensional Markovian latents connected by deep neural networks and conditioned on a discrete latent as a state-space embedding for temporal weights to account for nonlinear dynamics, enable data clustering in a low dimensional space, and provide informative visualizations about data. To tackle high spatial dimensionality in fMRI, we employed a low dimensional spatial embedding to generatively parameterize spatial factors. DMFA proves fast and capable on large-scale fMRI data.

## Acknowledgment

This work was supported by award numbers 1944964 and 1835309 from the national science foundation (NSF). The authors would also like to thank the anonymous reviewers for their insightful feedback.

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
