## A More Visualizations for Autism Dataset

We have visualized real and reconstructed brain images from the nine subsets of autism dataset (Caltech, Leuven, MaxMun, NYU, SBL, Stanford, UM, USM, and Yale sites) along with $z_0^{\mathrm{w}}$ after training DMFA on the full autism dataset, in Fig. S1. DMFA clustered together temporal latent variables associated with each imaging site without supervision in $z_0^{\mathrm{w}}$. As depicted, the variation among different imaging sites dominates the cognitive differences between ASD group and control, hence, a downstream connectivity matrix classification (using the learned temporal weights, $W$) helps better in differentiating ASD group from control in multi-site analysis.

## B Hyperparameters for Real Data

We chose $K = 100$ in real experiments to make it roughly at par with the number of regions in the AAL atlas. In practice, $K$ should be selected by a practitioner depending on the desired granularity. We set the size of temporal latents $z_t^w$ to be 2-dimensional as we found it helpful for visualization purposes and because larger sizes had marginal effect on the results. We selected the size of the hidden layer for the temporal transition network $D_t$ from {2, 3, 5}, and picked the model with highest prediction accuracy on a small validation set. We set the size of the hidden layer for the temporal emission network $D_e$ to the geometric mean between the size of $z_t^w$ and $K$ (i.e., size of $w_t$), which is $D_e = \sqrt{2 \times 100} \sim 15$.

## C Background on TFA Models

TFA approximates each fMRI sequence $Y_n$ as a product between temporal weights $W_n = [w_{n,k,t}]$ and spatial factors $F_n = [f_{n,k,v}]$, and defines a hierarchical Gaussian prior over each of these latent variables:

$$Y_n \sim \mathrm{Norm}\left(W_n^\top F_n, \sigma^Y \mathrm{I}\right),$$

$$w_{n,k,t} \sim \mathrm{Norm}(\mu_{n,k}^w, \sigma_{n,k}^w), \qquad \mu_{n,k}^w \sim p(\mu^w), \qquad \sigma_{n,k}^w \sim p(\sigma^w),$$

$$f_{n,k,v} = \mathrm{RBF}(v; \rho_{n,k}, \gamma_{n,k}), \qquad \rho_{n,k} \sim p(\rho), \qquad \gamma_{n,k} \sim p(\gamma).$$

TFA treats each fMRI sequence as independent. HTFA works similarly to TFA, but places an additional constraint over the factors to bias all of the sequences to exhibit similar factors:

$$\mu_{n,k}^w \sim p(\mu_{n,k}^w \mid \bar{\mu}_k^w), \qquad \bar{\mu}_k^w \sim p(\bar{\mu}^w), \qquad \sigma_{n,k}^w \sim p(\sigma_{n,k}^w \mid \bar{\sigma}_k^w), \qquad \bar{\sigma}_k^w \sim p(\bar{\sigma}^w),$$

$$\rho_{n,k} \sim p(\rho_{n,k} \mid \bar{\rho}_k), \qquad \bar{\rho}_k \sim p(\bar{\rho}), \qquad \gamma_{n,k} \sim p(\gamma_{n,k} \mid \bar{\gamma}_k), \qquad \bar{\gamma}_k \sim p(\bar{\gamma}).$$

In this way, whereas TFA attempts to find the factors that best explain an individual sequence, HTFA assumes that the factors across sequences vary around a shared Gaussian prior. NTFA extends HTFA for task fMRI by incorporating neural networks onto its framework. NTFA assumes separate latent embeddings for participants and stimuli and from there maps into the temporal and spatial latents with neural networks. However, it is evident from the priors in TFA methods that the temporal weights are independent as a function of time and therefore these models do not encode temporal dynamics.

## D Likelihood Approximation

The model's likelihood is intractable, however, we computed an importance sampling based lower-bound approximation of the likelihood which is also adopted in Krishnan et al. [2017], Kingma and Welling [2014]. Let's denote $Y$ as the train data and $\bar{Y}$ as the test data. We approximated the test set posterior-predictive likelihood from $L = 100$ samples as follows:

$$\log p(\bar{Y} \mid Y) \geq \mathbb{E}[\bar{\mathcal{L}}] \approx \frac{1}{L} \sum_l \sum_t \log p_\theta(\bar{Y}_t \mid \bar{w}_t^{(l)}, \bar{\rho}^{(l)}, \bar{\gamma}^{(l)}, \bar{z}_t^{\mathrm{w}(l)}, z_{t-1}^{\mathrm{w}(l)}, z^{\mathrm{F}(l)}, \mathrm{C}^{(l)}), \qquad (1)$$

where superscript $(l)$ indexes a sample. We sampled $z_{t-1}^{\mathrm{w}}$, $z^{\mathrm{F}}$, and C from their variational distributions and the remaining latent variables from their priors, i.e., the generative model:

$$z_{t-1}^{\mathrm{w}(l)} \sim q(z_{t-1}^{\mathrm{w}}), \qquad z^{\mathrm{F}(l)} \sim q(z^{\mathrm{F}}), \qquad \mathrm{C}^{(l)} \sim q(\mathrm{C}),$$

$$\bar{z}_t^{\mathrm{w}(l)} \sim p(\bar{z}_t^{\mathrm{w}} \mid z_{t-1}^{\mathrm{w}(l)}), \qquad \bar{w}_t^{(l)} \sim p(\bar{w}_t \mid \bar{z}_t^{\mathrm{w}(l)}), \qquad \bar{\rho}^{(l)}, \bar{\gamma}^{(l)} \sim p(\bar{\rho}, \bar{\gamma} \mid z^{\mathrm{F}(l)}).$$

The likelihood is estimated per subject and the average is reported in Table 2 of the paper.

**Algorithm S1** DMFA Generative Model

---

1: **for** $n$ **in** $1, \ldots, N$ **do**
2:     $z_n^{\mathrm{F}} \sim \mathrm{Norm}(0, \mathrm{I})$
3:     $\rho_{n,1}, \ldots, \rho_{n,K}, \gamma_{n,1}, \ldots, \gamma_{n,K} \sim \mathrm{Norm}(\mu_\theta^{\mathrm{F}}(z_n^{\mathrm{F}}), \sigma_\theta^{\mathrm{F}}(z_n^{\mathrm{F}}))$
4:     **for** $k$ **in** $1, \ldots, K$ **do**
5:         **for** $v$ **in** $1, \ldots, V$ **do**
6:             $f_{n,k,v} \leftarrow \mathrm{RBF}(v; \rho_{n,k}, \gamma_{n,k})$
7:     $F_n \leftarrow [f_{n,k,v}]$
8:     $\mathrm{C}_n \sim \mathrm{Cat}(\boldsymbol{\pi}), \quad \boldsymbol{\pi} = \{\pi_1, \ldots, \pi_S\}$
9:     $z_{n,0}^{\mathrm{W}} \sim \mathrm{Norm}(\mu_{\mathrm{c}_n}, \Sigma_{\mathrm{c}_n})$
10:     **for** $t$ **in** $1, \ldots, T$ **do**
11:         $z_{n,t}^{\mathrm{W}} \sim \mathrm{Norm}\big(\mu_\theta^{\mathrm{Z}}(z_{n,t-1}^{\mathrm{W}}), \sigma_\theta^{\mathrm{Z}}(z_{n,t-1}^{\mathrm{W}})\big)$
12:         $w_{n,t} \sim \mathrm{Norm}\big(\mu_\theta^{\mathrm{W}}(z_{n,t}^{\mathrm{W}}), \sigma_\theta^{\mathrm{W}}(z_{n,t}^{\mathrm{W}})\big)$
13:     $W_n \leftarrow [w_{n,t}]$
14:     $Y_n \sim \mathrm{Norm}(W_n^\top F_n, \sigma^{\mathrm{Y}} \mathrm{I})$

---

# E  DMFA Generative Algorithm

We have summarized the generative model of DMFA in Algorithm S1.

# F  Autism Separation Accuracy

Table S1 includes the autism separation accuracy for all the subsets of autism dataset and all the methods.

# G  Sensitivity Analysis

Table S2 provides test set prediction errors for $D_t \in \{2, 3, 5\}$ and $\sigma^{\mathrm{Y}} \in \{0.001, 0.005, 0.01\}$ on different datasets. DMFA proves stable w.r.t. to these hyperparameters. Note that $D_t$ is the dimension of hidden layer for the temporal transition network and $\sigma^{\mathrm{Y}}$ is the observation noise.

# H  Societal Impact

Analysing brain imaging scans are intended for research aimed at understanding the pathophysiology of neurodegenerative disease and the development of treatments for use in the presymptomatic phase. However, an important neuroethical issue is the predictive and diagnostic imaging for progressive diseases that lack effective treatments, such as Alzheimer's disease. These imaging scans could be used for other reasons by the worried employers or insurers. In such cases, the benefits of foreknowledge, for example the greater opportunity to plan, must be weighed against the psychological burden of this knowledge and its potential impact on employability or insurability.

Table S1: Comparison of autism separation accuracy (%).

| Method \ Dataset | Caltech | Leuven | MaxMun | NYU | SBL | Stanford | UM_1 | USM | Yale | Full |
|---|---|---|---|---|---|---|---|---|---|---|
| DMFA | **70** | 50 | 57 | 52 | 62 | 71 | 59 | 51 | **56** | 51 |
| DMFA+SVM | 48±6 | **55±5** | **60±6** | 59±3 | **63±7** | **77±6** | 66±4 | 65±6 | 52±6 | **64±2** |
| ROI+SVM | 42±8 | 54±6 | 54±7 | **66±3** | 46±9 | 63±8 | **67±4** | **67±6** | 47±7 | 60±2 |
| ICA+SVM | 43±8 | 54±6 | 55±8 | **66±3** | 45±9 | 64±7 | 66±4 | **67±6** | 48±8 | 59±3 |

The best results are highlighted in bold fonts.

Table S2: Prediction error (NRMSE%) of DMFA versus $D_t$ and $\sigma^{\text{Y}}$ on the test sets.

| Hyperparameter
Dataset | $D_t = 2$ | $D_t = 3$ | $D_t = 5$ | $\sigma^{\text{Y}} = 0.001$ | $\sigma^{\text{Y}} = 0.005$ | $\sigma^{\text{Y}} = 0.01$ |
|---|---|---|---|---|---|---|
| Autism (Caltech) | 5.75 | 5.39 | 5.44 | 5.32 | 5.41 | 5.44 |
| Depression | 6.50 | 6.48 | 6.35 | 6.35 | 6.43 | 6.78 |
| Synthetic | 2.07 | 2.12 | 2.28 | 2.06 | 2.09 | 2.07 |

In other cases, brain imaging analysis raises new ethical, legal, and social issues that stem directly from the special relationship between brain and mind. The ability of brain imaging to deliver information about our psyches—about who we are and what we might be thinking or feeling while in the scanner—opens up a range of ethical challenges with few, if any, direct precedents. In other words, to the extent that brain imaging can actually deliver useful information about a person's mental states or traits, the issue of privacy gets important. To the extent that it cannot, but people believe that it can, the issue of public misunderstanding gets important. On the face of things, brain imaging poses a novel challenge to privacy in that it can in principle deliver information about thoughts, attitudes, beliefs, and traits even when someone offers no behavioral responses. On the other hand, studies suggest that laypersons may attribute greater objectivity and certainty to brain images than to other types of information about the human mind. This may contribute to the premature commercialization of brain imaging for various real-world applications [Farah, 2012].

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

Figure S1: **Real and reconstructed** brain images from the nine subsets of Autism dataset (Caltech, Leuven, MaxMun, NYU, SBL, Stanford, UM, USM, and Yale sites) showing the smoothing given by sparse factorization. **Visualizing** $z_0^w$ after training DMFA on the full autism dataset. DMFA clustered together temporal latent variables associated with each acquisition site without supervision. As depicted, the variation among different imaging sites dominates the variation in cognitive state of the brain (ASD group vs. control), hence, a downstream connectivity matrix classification helps better in differentiating ASD group from control in multi-site analysis.