# OpenReview forum: "Deep Markov Factor Analysis: Towards Concurrent Temporal and Spatial Analysis of fMRI Data"
_NeurIPS.cc/2021/Conference — NeurIPS 2021 Poster_

### Official Review · Reviewer_1fHu · 2021-06-25

**Rating:** 5
**Confidence:** 3

**Summary:**

The paper Deep Markov Factor Analysis (DMFA): Towards Concurrent Temporal and Spatial Analysis of fMRI Data proposes a scalable approach to factorize the data of each subject into a product of a small number of spatial and temporal components.
Following on the work of (Manning et al., Topographic Factor Analysis, Plos ONE 2014) on topographical factor analysis, each spatial component is given by a radial basis function. The temporal components are modeled from a set of latent satisfying the Markov property. The first latent is conditioned on the data class giving to the model clustering capabilities. Variational inference is used to derive a lower bound of the log-likelihood which is then optimized via stochastic gradient descent.
In a first experiment on resting state data (Autism dataset), DMFA is used to obtain subject specific atlases that allow to seggregate ASD and control subjects with an accutacy of 60 % (it is definitely hard to read).
On a second experiment (Depression dataset), MDD and controls are exposed to different musical stimuli and DMFA is able to seggregate well the data of different subjects without supervision. In some extend it gives some hint about the kind of stimuli and the conditions of the subjects but results are not so convincing (the authors are honest about this).
In a third experiment they show that DMFA gives better results than other methods in terms of held-out log-likelihood and prediction of the next time point given the history.
The code for DMFA is available. Pre-trained model are given and the code to generate the figures from pre-trained model works.


**Limitations And Societal Impact:**

Authors have a broader impact section in appendix.

**Main Review:**

# 1. Originality
The model proposed by the authors uses the same spatial prior as in (Manning et al., Topographic Factor Analysis, PloS one 2014) and a temporal prior similar to (Krishnan et al. Deep kalman filters, 2015.).
The combination of these two techniques is new.
I find that the related work section is well done overall and I find that the difference with previous work is clearly stated. In the section "Factor Analysis in fMRI" I would have added techniques related to IVA (Lee JH et al. Independent vector analysis (IVA): multivariate approach for fMRI group study., Neuroimage 2008), MultiViewICA (Richard Hugo et al., Modeling Shared responses in Neuroimaging Studies through MultiView ICA, NeurIPS 2020) or TensorICA (Beckmann et al., Neuroimage, 2005).

# 2. Quality
## 2.1 Sensitivity analysis needed
The proposed method has many hyper-parameters to tune (number and size of hidden layers, size of latent factors,  number of spatial maps to cite just a few). Some of these parameters vary from one experiment to the next such as  the size of hidden layers between synthetic and real experiments or the value of $\sigma_0$ between the autism and depression dataset.
Ideally, it would be nice to have sensitivity curves for all hyper-parameters but the fact that there are no sensitivity curves for the hyper-parameters that vary across experiments is problematic I think.

## 2.2 Autism experiment
1. Unless I missed something, the model is fit on the full dataset (train and test set) so that the time-courses of interest can be recovered in all subjects. I find this worrying because there is leakage of information from the test set. Ideally, the parameters of the model would be learnt on the train data and fixed when the test data are used. However with DMFA this is not possible because there is one set of parameters per subjects. I suggest here another experiment using naturalistic stimuli that I believe yields a cleaner pipeline: the authors could use the forrest dataset where subjects listen to an audio version of forrest gump (Hanke et al., Scientific data, 2014) to learn the spatial representations. Then they could use the second part of the dataset where the same subjects are listening to music of different genre and attempt to recover the music genre from the projected data.
2. I find it quite surprising to use temporal ICA sources to build a functional connectivity matrix. These sources are typically independent and in most ICA algorithms, they are normalized so the functional connectivity matrix should always be identity unless I am wrong. Maybe it makes more sense to use spatial ICA sources to get the atlases and then project the data onto these atlases ?
3. A standard baseline would be to use the 444 regions of the BASC Atlas (Bellec, Multi-level bootstrap analysis of stable clusters in resting-state fmri., NeuroImage 2010) to get connectivity matrices before SVM is applied on them (with python the atlas is easy to get using nilearn https://nilearn.github.io/modules/generated/nilearn.datasets.fetch_atlas_basc_multiscale_2015.html ). I fill it is a more natural baseline than the ones tried by the authors.
4. The result for DMFA (without SVM) are missing when the full dataset is used. Why did the authors make this choice ?
5. I would have expected to see the results of competitive unsupervised methods (NTFA, HTFA) as well as the ablation study done here so that we can compare the methods based on the decoding accuracy they yield.
6. The authors state that competitive methods for the same task reach 69 % accuracy (l256). I believe it would be fair to display this information in Figure 3 so that we can actually see where DMFA stands (I believe DMFA is below 69 % isn't it ?)

## 2.3 Depression experiment
There is clear evidence that the different subjects are in different clusters. However, what is interesting here in my opinion is whether the model can retrieve the condition of the different subjects (or some characteristics of the displayed stimuli) better than alternatives. About the condition of subjects authors say: (l284) "MDD group data points are fairly populated in the center of temporal latent while control group are dispersed across latent space". I find this statement a bit fuzzy. As it is, this experiment does not allow me to formulate a clear conclusion about whether the model works better than alternatives or not.

## 2.4 Comparison with state of the art
I am not sure I fully understood this part.
1. This comment asks for more pedagogy. The likelihood is intractable. How can the authors evaluate it ?
2. Leaving aside the problem of information leakage from the test set, I think the previous experiments on task fMRI or rest fMRI data provide more classical benchmark in neuroscience and more importantly yield results that are relevant in clinical applications (if a method can classify better than alternatives whether a patient is ASD or not this would really yield practical benefits).  Although the held-out likelihood / prediction of next sample suggests that DMFA is a better generative model, I find it hard to know whether these metrics are really reliable and if it is of any use in neuroscience.

# 3. Clarity

I find this submission extremely difficult to follow.

## 3.1 Notation and organisation issues
1. The notations become clearer in Table 1 but when reading section 3, I don't know the dimensions of the latent variables $z_n^F$, the domain and co-domain of function $\mu_{\theta}^W$ etc. The space in which any parameter leaves should be defined straight away at least in the text if it is not done directly in the formulas. This is really an important point as it makes the submission difficult to understand.
2. The notations $\mathcal{N}$ and $Norm$ are redundant unless I missed something. All of these notations should be properly defined in a notation section.
3.  Are $\sigma_{\theta}^F$ matrices ? I believe so but I am not sure. Matrices, vectors and reals should have a different notation (using uppercase, bold and lowercase for example). Another example  l132 $g$ lives in $[0, 1]$ but in Table 1 it lives in $[0, 1]^2$. I understand these are typos but with clearer notations, it becomes obvious to know what is correct.
4. I believe $\sigma_0$ is never defined.

## 3.2 Motives for modelization choices
1. Having a common name for groups of parameters such as $\theta$ or $\phi$ may be practical but I find it confusing in the sense that it hides the fact that some parameters are shared and some are not. In general it is not always clear which parameters are shared or not. The notations give some hints but it would be better if this was explicitely stated in the text. In addition, I believe it is important to understand why these choices are made. For example, you assume that the spatial components are subject specific as well as the temporal components. These assumptions are not made in all models (for example in (Calhoun, A method for making group inferences from functional MRI data using independent component analysis.,  Human Brain Mapping 2001) the temporal part is assumed to be shared across views). Other parameters are shared (such as the neural network functions) and we might want to know why.
2. Table 1 describes a particular architecture. It would be nice to understand how the authors end up chosing this particular one and what is the performance of other choices.

## 3.3 Readability
1. The figures are too small especially Figure 3 (right) or the legend in Figure 4 (a). In Figure 3 (right) this is particularly problematic as it is difficult to read the accuracy values. A grid might also help.
2. Table 1 is difficult to understand. There is this layer column which goes down to 10 layers whereas if I understand correctly, none of the networks have more than 1 hidden layer.
3. In order to facilitate the reading of the derivation of the ELBO it would have been nice to see a more detailed version in appendix.
4. (l309) "trained each with their default hyperparameter": what are the defaults hyper-parameters for DMFA ? For other models ? According to which paper or which package ?

## 3.4 Reproducibility
1. l226 "These three groups of sources are periodically activated according to some random weights". The type of randomness should be precised.
2. In the depression dataset there is no value for $S$.
3. I find it difficult to reproduce the figures from scratch using the code as it involves downloading the full dataset and pre-processing it myself (in the README there is no script to get the dataset although there seems to be one to do the pre-processing).  What I would have hoped for is a small synthetic experiment that I can run in a short amount of time and is just there to show me that the code works.

# 4 Significance
1. Ignoring the fact that some information may leak from the test set, results displayed in Figure 3 are only marginally better than SVM after applying an atlas. For a method that takes 12 000 minutes to run, I think we need a marked improvement if we want other researchers to use it.
2.  However, I would like to highlight that with fMRI data it is quite difficult to study the temporal structure when the TR is as high as 2s. I would suggest the authors to take datasets where the TR is much lower such as (Feinberg, et al. "Multiplexed echo planar imaging for sub-second whole brain FMRI and fast diffusion imaging." PloS one 2010) so that they have a higher chance of seeing strong effects.


# 5 Minor comments
l157 The comment that amortized variational inference could lead to over-fitting is wrong in my opinion. Amortized variational inference is strictly less powerful than variational inference as in amortized variational inference the parameters are restricted to leave in the image of the parametrized function while in variational inference they are completely free.

**Time Spent Reviewing:**

15

---

> ### Author Response · Authors · 2021-08-10
> **Thank you.**
>
> Thanks for your detailed and valuable feedback.
>
> Thank you for introducing references of IVA (Lee JH et al.), MultiViewICA (Richard Hugo et al) and TensorICA (Beckmann et al). We added a discussion of these methods to our related works section.
>
> **Information leakage from test set?**
>
> We believe that there is a slight confusion here. For reporting the results in Table 2 (i.e., test set log-likelihood and test set prediction), we split the data into train and test, and fit the model on the train data to learn parameters of the generative model ($\theta$) and variational parameters ($\phi$) of the train data. Then, we fixed all parameters of the generative model, (i.e., neural network mappings of $\mu_\theta^z$, $\sigma_\theta^z$, $\mu_\theta^w$, $\sigma_\theta^w$, $\mu_\theta^F$, $\sigma_\theta^F$, and $\mu_s$, $\Sigma_s$), and run inference on the test set to estimate variational parameters of the test set (i.e., parameters of their Gaussians $q_\phi(Z^w)$, $q_\phi(z^F)$, $q_\phi(W)$, $q_\phi(\rho, \gamma)$ denoted by $\phi$) using the generative model learned on the train set. This is done by running the model for a few iterations on the test set (for test set variational parameters to converge) while generative parameters are fixed (not updated). This is the standard approach employed for non-amortized inference which is also used in NTFA. For an amortized inference (e.g., in VAE), this is done by calling the forward pass of the encoder on the test set to estimate test set variational parameters.
>
> **Functional connectivity matrix from ICA time courses?**
>
> Please note that we did not perform temporal ICA. We applied spatial ICA to the dataset: $Y = W F$, where $Y\in R^{T\times V}$ (each column represents a single voxel time course), and $W\in R^{T\times K}$ is a matrix of $K$ time courses (TCs) and $F\in R^{K\times V}$ is matrix of $K$ spatial maps which are pairwise independent in a statistical sense. In other words, spatial ICA solves the problem of finding matrix $W$ (TCs or mixing matrix) such that $Y=WF$ and rows of $F$ are pairwise independent (Note that matrix $W$ is unconstrained). We computed correlation matrix (functional connectivity matrix) between columns of $W$. Kindly see section 2 in https://www.fmrib.ox.ac.uk/datasets/techrep/tr01mj2/tr01mj2.pdf, and sections 2.2.3. and 2.2.4. in https://www.ncbi.nlm.nih.gov/pmc/articles/PMC6438914/ which computed connectivity matrix of TCs (temporal courses) estimated from a spatial group ICA and used them for separating schizophrenia from control by applying SVM on them.
>
> For nonlinear ICA of Hyvarioan et al. considering that $F=[F_1, …, F_V]$ where $F_v$ represents its $v^\text{th}$ column, we have $Y = W \big[f(F_1), …, f(F_V)\big]$, where $f(.): R^K \rightarrow R^K$ is a nonlinear feature extractor (e.g., an MLP) applied to each column of $F$ and is learned, and rows of $F$ are forced to be independent (while $W$ is unconstrained). We computed correlation matrix from $W$ (the $K$ time courses in the mixing matrix). Kindly see https://arxiv.org/pdf/1605.06336.pdf sections 3 and 4, and equation 1 and 3 and corollary 1 for problem formulations.
>
> **DMFA (without SVM) results on the full Autism dataset**
>
> We have visualized $z_0^w$ (first temporal latent) after training DMFA on the full Autism dataset in Fig. S1 bottom-right (see supplementary). Note that on the full dataset, DMFA is not able to cluster ASD from control in an unsupervised way, because, as shown in this figure, the variation among different imaging sites dominates the variation in cognitive state of the brain (i.e., ASD group vs. control), therefore, a downstream connectivity matrix classification (as opposed to a clustering method) helps better in differentiating ASD group from control in this multi-site analysis. Kindly note that for this multi-site dataset, classification methods hardly achieve above chance accuracies, therefore it’s not surprising for a clustering method to fail.
>
> **DMFA+SVM accuracy on the full Autism dataset**
>
> DMFA achieved an average accuracy of 64%. Please note that accuracy of 69% is achieved on a specific split of train/test data to minimize site-related variability and make data as homogeneous as possible. We have observed DMFA+SVM to achieve 72% on a specific train/test split of this dataset.
>
> Kindly note that DMFA provides additional information about fMRI data compared to classical methods (e.g., subject-level variability in temporal latents) and at the same time enables classical fMRI analysis by providing the weights and factors (e.g., downstream regression/classification tasks). Although DMFA gives comparable results on some subsets of Autism dataset in terms of classification accuracy, it provides visual hints about the arrangement of data in 2-dimensional latents which were not directly possible using classical methods.
>
> **Results of HTFA/NTFA for Autism classification**
>
> We followed up and performed classification on the connectivity matrices obtained from HTFA. The accuracy results are comparable to that of ROI+SVM. We will include this and comparison results with NTFA in the revised version of our paper.
>
> **Computing likelihood?**
>
> The likelihood is intractable. However, we compute an importance sampling based lower-bound approximation of the likelihood which is also adopted in NTFA (Sennesh et. al., NeurIPS'21, see their section 4.3 and equation 11), DMM (Krishnan et.al., AAAI’17, https://arxiv.org/pdf/1609.09869.pdf, see equations 9, 10 in their Appendix A), and Auto-encoding variational bayes (Kingma and Welling, ICLR’14, Appendix D).
>
> In brief, assuming $\tilde{Y}$ to be a test data and $Y$ to be the train data, we approximate posterior-predictive likelihood from $L=100$ samples (superscript $(l)$ denotes a sample):
>
> $$p(\tilde{Y}|Y) \geq E[\tilde{\mathcal{L}}] =\frac{1}{L}\sum_l \sum_t \log p_\theta(\tilde{Y}_t | \tilde{w}_t^{(l)}, \tilde{\rho}^{(l)}, \tilde{\gamma}^{(l)}, \tilde{z}_t^{w(l)}, \tilde{z} _{t-1}^{w(l)}, z^{F(l)}, C^{(l)}),$$
>
> We sample $z^w_{t-1}$, $z^F$, and $C$ from their variational distribution and the remaining latent variables from their prior (i.e., generative model):
>
> * $\tilde{z}^{w(l)}_{t-1}\sim q(\tilde{z}^w _{t-1}),\quad z^{F(l)} \sim q(z^F),\quad C^{(l)} \sim q(C)$
>
> * $\tilde{\rho}^{(l)}, \tilde{\gamma}^{(l)} \sim p(\tilde{\rho}, \tilde{\gamma}|z^{F(l)}), \quad \tilde{z}_t^{w(l)} \sim p(\tilde{z}_t^w | \tilde{z}^{w(l)} _{t-1}), \quad \tilde{w}_t^{(l)} \sim p(\tilde{w}_t|\tilde{z}_t^{w(l)}).$
>
> \
> **Significance**
>
> Please note that our method takes $200\times 6$ min = $1200$ minutes on the entire Autism dataset (**not** $12000$ minutes). The Autism dataset is a huge dataset (around 1000 fMRI data) and 200 epochs are more than enough for convergence.
>
> Thank you for suggesting the dataset of Feinberg, et al., we will explore that.
>
> **Motives for modeling choices**
>
> The neural network mappings in the generative model are shared across all data points (i.e., all subjects) such that they define a generative model over the entire dataset. It is necessary to have temporal components (mixing weights of spatial components) vary over time and among subjects, because brain activations are independent across subjects at a specific time point especially in the case of rest-fMRI. We chose spatial components to slightly vary across subjects to capture subject-level variability in this matter. This is done by defining a prior over spatial components (locations and extents of Gaussian blobs) which allows these components to slightly perturb across data points.
>
> **Hyperparameter selection and sensitivity analysis**
>
> In general, hyperparameters can be set according to the model’s performance in terms of held-out likelihood or prediction accuracy on a validation set.\
> Here, we chose $K=100$ in real experiments to make it roughly at par with the number of regions in AAL atlas. In practice, $K$ should be selected by a practitioner depending on the desired granularity.
> We set the size of hidden latents $z^w_t$ to be $2$ as we found it helpful for visualization purposes and because larger sizes only had a slight effect on the results.
> We selected the size of the hidden layer for the temporal transition network (i.e., $D_t$ in Table 2) from {$2,3,5$}, and picked the model with highest prediction accuracy on a small validation set ($D_t = 5$ was selected here). We set the size of the hidden layer for the temporal emission network (i.e., $D_e$ in Table 2) to the geometric mean between size of $z^w_t$ (i.e., $2$) and $K$ (size of $w_t$), which is $D_e = \sqrt{2\times 100} \sim 15$.
>
> We will report sensitivity curves for $D_t$ and $\sigma_0$ in the supplementary.
>
> **Editing comments**
>
> Thank you for suggesting these edits. We will follow these rules for our notations and continue editing our paper to improve the presentation of our work.
>
> **Reproducibility**
>
> For synthetic data, we sampled from a uniform distribution between -½ and ½.\
> In the depression dataset $S$ is set to $19+20=39$ ($39$ subjects in total).
>
>
> **Comment on amortized inference**
>
> Without proper choice of an encoding model, amortized inference can lead to overfitting when the feature dimension of data is much higher than the number of data points. This is often the case in fMRI data as we have only a few subjects (i.e., data points) and thousands of voxels (i.e., features). For example a simple RNN-MLP encoder on the vectorized fMRI data results in $O(VK)$ parameters for amortized inference while non-amortized inference needs $O(NTK)$ parameters, and for fMRI data we often have $N\sim 50$, $T\sim 100$, while $V\sim 200,000$, therefore $VK \gg NTK$. However, by using a RNN-CNN encoder (e.g., with Conv3d for 3D MRI images) an amortized inference may need fewer parameters than its non-amortized version, but Conv3d is computationally very expensive and demands intense GPU memory consumption and results in very long training times.

---

> > ### Comment · Reviewer_1fHu · 2021-08-26
> > **Thanks for your response.**
> >
> > 1. Information leakage from the test set
> >
> > I was referring to the autism experiment. Unless I misunderstood you use all subjects to fit the model including test subjects.
> > After reading your comment about the results of Table 2 I still think that using the test set to estimate variational parameters is a bad practice. I understand that with your model there is no other choice but to me this is a strong weakness of the model. If you were to compare your model with any other classical factor model such as PCA it would be so unfair: because you use the test set to estimate variational parameters, you are basically insensitive to any distribution shift whereas PCA would be extremely sensitive to this.
> > To be clear, I think that an issue with the model is that you have one variational parameter per sample. Otherwise, you would be able to just fix all parameters when you apply your method on the test set.
> >
> > 2. temporal / spatial ICA
> >
> > I was refering to l253 "temporal sources obtained with nonlinear ICA". So you actually meant "spatial sources" here. By the way, I believe it is more standard in fMRI to use just spatial ICA instead of the non-linear extensions.
> >
> > 3. Results of HTFA/NTFA for Autism classification
> >
> > Great. This makes it more convincing.
> >
> > 4. Computing likelihood?
> >
> > I believe this makes it not the best metric then, we have no reason to believe that the lower bound is tight.
> >
> > 5. Significance
> >
> > Sorry for my mistake. But I believe the point is still valid. Unless I am wrong, 1200 minutes is still much much larger than competitive linear methods.
> > I appreciate the work done in the rebuttal but I still believe the paper is Ok but not good enough for NeurIPS.

---

> > > ### Author Response · Authors · 2021-08-27
> > > **Thanks again for your feedback.**
> > >
> > > Thanks again for your time in responding to our rebuttal.
> > >
> > > ## 1) Estimating test set variational parameters from the test set is not a weakness of the model, it is a necessity:
> > > Kindly note that using the test set to estimate *test set variational parameters* is not a modeling choice, it is a necessity. Different methods use the test set in different ways to estimate corresponding posterior parameters. In the following, we explain how the test set is used for estimating its posterior parameters in variational inference, amortized inference, and PCA/ICA:
> > >
> > > * In standard variational inference, local test set variational parameters (e.g., $\mu$, $\sigma$ of Gaussian distributions) are estimated from the test set as free parameters while parameters of the generative model are *fixed*. Note that the goal here was to learn a generative model from the training data.
> > >
> > > * In amortized variational inference, test set variational parameters are estimated by passing the test set to an encoder: e.g., $\mu_{\tilde{X}}, \sigma_{\tilde{X}} = f(\tilde{X})$, where $\tilde{X}$ is the test data and $f(.)$ is a mapping (a.k.a. inference model). The amortized inference was introduced to make variational inference scalable to large $N$ (data points) by learning a single parametric function $f(⋅)$ which basically maps a given data (train/test) to a set of variational parameters tailored to that datapoint (instead of defining local variational parameters). And the goal is to learn an inference model (encoder) in addition to the generative model.
> > >
> > > However, for small to medium-sized $N$, using local variational parameters can be a better approach (as we explained in our previous response) because the amortized inference is not computationally efficient for our case.
> > > These concepts are perfectly explained in this short document: https://pyro.ai/examples/svi_part_ii.html#Amortization
> > >
> > > * For PCA/ICA, after fitting these models on a training dataset, $Y$, we obtain temporal and spatial factors: $Y= W F$, where $W\in R^{T\times K}$ and $F\in R^{K\times V}$. And $F$ are the basis of their generative model. For a test data $\tilde{Y}$, its temporal weights $\tilde{W}$ are estimated from $F$ by solving the following equation:
> > > $$\tilde{Y} = \tilde{W} F  \rightarrow \text{(least-squares solution)}  \tilde{W} = \tilde{Y} F^{\dagger}$$
> > > where $F^{\dagger}$ is the Moore–Penrose inverse of $F$. For an analogy, here $F$ serves as the basis of generative model (learned on train data) and $\tilde{W}$ are posterior parameters corresponding to test set $\tilde{Y}$.
> > >
> > > **It is clear that in all of these cases test set is indeed used for estimating corresponding variational/posterior parameters, either through some free local parameters, or by passing the test set to an inference model (encoder), or by solving a least-squares solution on the test set.**
> > >
> > > ## 2) We used temporal sources obtained with nonlinear spatial ICA:
> > >
> > > We emphasize that we did not use "spatial sources". We used temporal sources (i.e., time courses) obtained with nonlinear spatial ICA. When spatial ICA is applied to a dataset $Y\in R^{T\times V}$ (each column represents a single voxel time course), we obtain $F\in R^{K\times V}$ which is a matrix of $K$ spatial maps which are pairwise independent in a statistical sense, and $W\in R^{T\times K}$ which is a matrix of $K$ time courses (TCs) and are unconstrained. We computed the correlation matrix (functional connectivity matrix) between columns of $W$ (not $F$). This is similar to previous works as we explained in our previous response.
> > >
> > >  ## 3) Computing likelihood?
> > >
> > > We agree that this might not be the best metric, but since the true posterior is intractable, this measure is widely accepted as a comparison metric in the literature due to its theoretical justifications as there are few/no alternatives.
> > >
> > > The test set prediction error (which we used for comparison with RKN), however, is a solid measure for comparing dynamical models. We couldn't use this measure for HTFA and NTFA because these models are not predictive.
> > >
> > > ## 4) Group ICA is also very time-consuming. Our run-time is on a CPU
> > >
> > > A standard group ICA on 1000 fMRI data takes as much as our method. Please note that our dataset is very huge. Also, we wanted to emphasize that our run-time is reported on a CPU (not GPU). A GPU can speed up the process significantly.
> > > \
> > > \
> > > ####Update#####
> > >
> > > We really appreciate the reviewer's time in providing feedback.
> > >
> > > ### 1) The computational complexity of SVD for a subject is $O(T^2 V)$ (assuming $T<V$) and the computational complexity of ICA is $O(T^2 V m)$ where $m$ is the number of iterations needed for convergence (and in practice, it grows with data dimension)-- see Table I in https://isp.uv.es/papers/Laparra11.pdf
> > >
> > > On our best system (CPU Core i9 with 64 Gb RAM), a matrix of size 200 000 x 300 takes $13.54$ seconds for computing the top 300 SVD components (the highest rank of the covariance matrix) in MATLAB v2021 using the `svds` function (by the way, using `svd` function is not feasible and would return the following error: Requested 200000x200000 (298.0GB) array exceeds maximum array size preference. We are assuming that the reviewer is not referring to `svd` because it takes 10 minutes for 90 000 x 300, the largest we could fit into memory.)
> > >
> > > For ICA, considering 884 subjects and only 5 iterations for convergence! would give 13.54 x 884 x 5 $\approx$ 59846 seconds or 997 minutes in the best case, which is apparently in the range of DMFA. Please note that neural networks in DMFA are some MLPs in the lower dimensional space and can be optimized very efficiently.
> > >
> > > ### 2) Regarding spatial ICA
> > >
> > > Thank you. We will revise the sentence and replace "temporal sources" with "time courses (TCs)" which is a widely used terminology in the neuroscience community when referring to temporal components obtained from applying spatial ICA.

---

> > > > ### Comment · Reviewer_1fHu · 2021-08-27
> > > > **Thanks for the response**
> > > >
> > > > Thanks for your response
> > > > 1. Ok I believe my previous comment was building on a false intuition. Thanks for the clarification.
> > > >
> > > > 2. I believe there is a terminology issue from your end. When you say you extract temporal sources it means that the sources are temporal and since the independence assumption is on the sources it means that you are using temporal ICA.
> > > >
> > > > 3. Noted
> > > >
> > > > 4. GroupICA also takes 1200 minutes: I don't believe this is true. If you take the implementation of Calhoun 2001 you basically only need to do a PCA per subject. On my laptop, an svd of on subject (matrix of size 200 000 x 300) takes about 5 seconds so even if you have 1000 subjects, this should be a lot faster than 1200 minutes especially if you use a cluster (even a CPU one).

---

> > > > > ### Author Response · Authors · 2021-08-27
> > > > > **Thanks for your response**
> > > > >
> > > > > We really appreciate the reviewer's time in providing feedback.
> > > > >
> > > > > ### 1) The computational complexity of SVD for a subject is $O(T^2 V)$ (assuming $T<V$) and the computational complexity of ICA is $O(T^2 V m)$ where $m$ is the number of iterations needed for convergence (and in practice, it grows with data dimension)-- see Table I in https://isp.uv.es/papers/Laparra11.pdf
> > > > >
> > > > > On our best system (CPU Core i9 with 64 Gb RAM), a matrix of size 200 000 x 300 takes $13.54$ seconds for computing the top 300 SVD components (the highest rank of the covariance matrix) in MATLAB v2021 using the `svds` function (by the way, using `svd` function is not feasible and would return the following error: Requested 200000x200000 (298.0GB) array exceeds maximum array size preference. We are assuming that the reviewer is not referring to `svd` because it takes 10 minutes for 90 000 x 300, the largest we could fit into memory.)
> > > > >
> > > > > For ICA, considering 884 subjects and only 5 iterations for convergence! would give 13.54 x 884 x 5 $\approx$ 59846 seconds or 997 minutes in the best case, which is apparently in the range of DMFA. Please note that neural networks in DMFA are some MLPs in the lower dimensional space and can be optimized very efficiently. As mentioned, using larger batch sizes (if memory permits) and GPU would benefit DMFA significantly (similarly, ICA can also be performed faster e.g., on cluster CPU).
> > > > >
> > > > > ### 2) Regarding spatial ICA
> > > > >
> > > > > Thank you. We will revise the sentence and replace "temporal sources" with "time courses (TCs)" which is a widely used terminology in the neuroscience community when referring to temporal components obtained from applying spatial ICA.

---

> > > > > > ### Comment · Reviewer_1fHu · 2021-08-30
> > > > > > **Thanks for your response**
> > > > > >
> > > > > > 1. I use python but if you use svd(A,'econ') in matlab you should get the same memory consumption and computation time than me (I would be surprised if matlab and python were not relying on the same C code to perform the SVD). Note that PCA is just an SVD performed on centered data. If you follow (Calhoun, 2001, A method for making group inferences from functional MRI data using independent component analysis) you would just do one PCA per subject then, take the reduced data, concatenate them, perform a second PCA and then the ICA. At the end of the day, if you want n_sources=100, you do the ICA on a matrix of size (n_sources, V) so it is very fast (it takes about 30 seconds on my laptop). In order for the second PCA to be scalable - and to handle memory issues - you need to chose a small number of components in the first PCA so you would typically use 10 components, then stack the data and perform the second PCA. You should get much lower runtime for the same performance.
> > > > > >
> > > > > > 2. Although I do believe that a fit time of 1200 minutes is a bit too high considering the improvements shown by the authors this was not my only point. Along with the fitting time issue, other major points to me are the clarity of the paper (notations and organization issues, the readability, the motives for modelling choice, the reproducibility issues) and the choice of non-standard baselines (most practitioners rely on already computed atlases such as BASC or use linear ICA: the performance of such methods along with their fitting time should be reported).
> > > > > >
> > > > > > I will however raise my score to 5 to reflect that some of my important concerns have been addressed.

---

### Official Review · Reviewer_RgVr · 2021-07-09

**Rating:** 6
**Confidence:** 4

**Summary:**

The paper proposes deep Markov factor analysis (DMFA) that uses the Markov process to capture the temporal dynamics in the fMRI dataset and maps the high spatial dimensions to a low-dimensional feature space. DMFA can cluster fMRI responses into “low dimensional temporal embedding” based on subject or cognitive state. The empirical studies on synthetic and real fMRI datasets illustrate that DMFA generates better performance in comparison with state-of-the-art techniques --- such as NTFA and HTFA.

**Limitations And Societal Impact:**

Please see the main comment section.


**Main Review:**

The following are the major concerns and minor comments:

1) My first question is about the prior and trail distribution(s). You can replace the combination of Normal distribution and RBF kernel with the Gaussian Process (Wu, 2019, J. Machine Learning Research) or even a Deep Gaussian Process (Roger Frigola-Alcalde, 2015, Thesis at University of Cambridge). The author(s) should explain why they chose these assumptions and how they can affect the performance of the analysis.

2) The proposed method is compared mostly with the same type of factor analysis. I would like to see comparisons with other paradigms of analyzing using similar datasets. For instance, comparing the performance of the proposed method with the graph-based approach (e.g., Mingliang Wang et al., 2021, Medical Image Analysis) or functional connectivity-based ones --- e.g., Lei Sun et al. 2021 Artificial Intelligence in Medicine.

3) The proposed method can be summarized in the form of an algorithm --- perhaps at the end of Section 3.

4) In this paper, the notations are confusing. In the regular papers, scalers are denoted by small letters, vectors are defined with small letters (highlighted by bold), matrices are denoted by capital letters using bold. In this paper, they are a lot of conflicts. It is so hard to trace what is a set, a matrix, or even a distribution.

5) The equations should have numbers. It is hard to trace them in the current format.

6) In this paper, "embedding" is used to describe the temporal low dimensional space. Mathematically, embedding spaces have greater dimensions than the original spaces (not lower). For clarity, I suggest the phrase "low-dimension feature space" be used here instead.

7) The brain images in Figures 2--4 and S1 should have value bars to make it clear what each color represents. Additionally, some subfigures do not have legends. The font size in these figures should also be increased.


**Time Spent Reviewing:**

30

---

> ### Author Response · Authors · 2021-08-10
> **Thank you.**
>
> Thanks for your valuable feedback.
>
> **Why not Gaussian process? Why variational inference?**
>
> The main computational goal for DMFA (and TFA methods) is to estimate the posterior distribution of the hidden variables given the data. In theory we could compute this posterior using Bayes’ rule:
> $$p(C, Z^w, z^F, W, \rho, \gamma|Y) = \frac{p(Y, C, Z^w, z^F, W, \rho, \gamma)}{p(Y)},$$
> where,
> $$p(Y) = \sum_C \int_{Z^w} \int_{z^F}\int_W \int_{\rho,\gamma} p(Y, C, Z^w, z^F, W, \rho, \gamma) \ dZ^w \ dz^F \ dW \ d\rho \ d\gamma$$
>
> However, computing $p(Y)$ is intractable, because it requires integrating over all possible combinations of values that the hidden variables could take on. (This is both analytically difficult and computationally intractable.) Therefore the posterior is intractable. This is the main reason for using approximate variational inference and mean-field approximation to break-down the computational complexity of the model's posterior. For the same reason, NTFA and other TFA methods also used black-box variational inference (BBVI).\
> Also please note that the hierarchical relationship between latents, together with nonlinear mappings that parameterize these conditional distributions (from the conditioning latent), and also the discrete latent break the Gaussianity.
> \
> \
> **Comparison with graph-based approach of Mingliang Wang et al., 2021 or functional connectivity-based approach of Lei Sun et al. 2021?**
>
> Thank you for introducing these baselines. Here, we give a brief description of these two papers to highlight their differences with DMFA.
>
> * The graph-based approach of Mingliang wang et.al, split rest-fMRI time sequences of each subject into several segments and compute functional connectivity matrix of each segment from some predefined ROIs in the brain. So, their data is a collection of connectivity matrices for each time segment and subject, and their associated label (patient vs control). They learn label predictors for each time segment, $\Theta_t$, to transform/map connectivity matrices to their associated labels (see their equation 3) and regularize these label predictors with $||\Theta_t - \Theta_{t-1}||_{1,2}$ such that subsequent predictors are close to each other and vary smoothly over time. They use majority voting between labels of segments to classify an fMRI data as control or patient.\
> In contrast to DMFA and TFA methods which process raw high-dimensional fMRI data for factor analysis, this method works with ROI data. In addition, this method does not encode/model temporal transitions of fMRI time series (or transition of connectivity matrices) and therefore is not predictive in this sense as DMFA (this method only learns a collection of predictors over time that are forced to be smooth.)\
> We will compare our method with theirs in terms of classification accuracy considering AAL atlas regions, and report that in the revised version of our paper.
>
> * The work of Lei Sun et.al proposes a new measure for computing functional connectivity matrix from ROI signals. Please note that we can also use this measure (instead of correlation) for computing functional connectivity matrices from the weight matrix (W) obtained from DMFA or other TFA/ROI methods.\
> We will report the classification results of this method in our paper. (From Table 2 of their paper, it appears that they could achieve average accuracies from 57.6% to 62.11% (based on different conditions) using AAL atlas regions and a 5-fold cross validation on the full Autism dataset.)
>
> **Summarizing DMFA in the form of an algorithm**
>
> Thanks for your suggestion. We will include an algorithm of DMFA in the supplementary.
>
> **Editing comments:**
>
> Thank you for suggesting these edits. We will follow these rules for our notations and continue editing our paper to improve the presentation of our work.
>
> We will replace embedding with “low-dimension feature space”.
>
> We will add color bars and legends to brain images and fix their font sizes.

---

### Official Review · Reviewer_m9hn · 2021-07-14

**Rating:** 6
**Confidence:** 4

**Summary:**

Extending topographical factor analysis (TFA), this paper proposes a model, deep Markov factor analysis (DMFA), that models temporal dynamics in embedding space. The model is applied to both simulated and real fMRI data.

**Ethical Concerns:**

There are no ethical issues in this paper.

**Limitations And Societal Impact:**

An elaborate discussion societal impact is provided in the paper. Limitation on implementation is also provided.

**Main Review:**

Originality

The proposed model is a novel extension of TFA with implementation details carefully worked out.


Quality

The method seems solid, but the results are not as impressive. A few comments below.
1. Is TFA always better than parcellating and averaging? Parcels are often generated from external datasets whereas latent representation are often built from datasets that are being analyzed. Would the former provide cleaner prediction evaluation?
2. How is using Gaussian blob more interpretable than spatial ICA, which tends to generate biologically interpretable networks?
3. The current simulation seems to be based on assumptions of the proposed model, and model hyperparameters are set to the ground truth. Also, quantitative comparison against TFA, HTFA, NTFA, and RKN should be provided.
4. How were hyperparameters selected for real data?
5. DMFA only beats ROI/ICA + SVM for some ABIDE datasets. Using p<0.02 as threshold would not survive multiple comparison correction for the number of datasets, i.e. 0.05/10 = 0.005.
6. What model is used for estimating likelihood in Table 2? Is likelihood estimated per subject? What is the standard deviation? If consider the errorbar, is DFMA’s Nats significantly lower than other methods?


Clarity

The paper is clear and concise. A few minor comments below.
1. Would ST-GCN fall under the scope of this paper? If so, please compare.
2. More description on TFA would be useful for readers to get a better sense of the mentioned limitations.


Significance

The model would be useful for modeling any spatiotemporal data, but the results do not convincingly show better performance over existing methods.


Post Author Responses

I have read through the other reviews and responses, and will keep my original score mainly since classification improvements are only seen for half of the tested datasets.

**Time Spent Reviewing:**

6

---

> ### Author Response · Authors · 2021-08-10
> **Thank you.**
>
> Thanks for your valuable feedback.
>
> Thanks for highlighting the novelty and solidity of our work.
>
> **Is TFA always better than parcellating and averaging?**
>
> Parcellating and averaging is an effective approach for functional connectivity analysis in fMRI studies as parcels can effectively be determined from structural information, and we are not claiming that TFA methods are always better than this approach. However, there are several limitations with this approach:
> * As mentioned, parcels are often obtained from external datasets and are assumed to be global across all subjects, so they smooth out intra-subject variability. By using Gaussian blobs of the same or more resolution (and by automatically learning their locations and extents) we let the model determine significant activation locations and also preserve subject-level variability. Please also note that an arbitrary shaped brain region can be roughly captured by superimposition of enough Gaussian blobs.
> * Also, there could be a case that a brain parcel includes multiple activation sources. This can be captured by multiple Gaussian blobs, but could be lost by averaging across the entire parcel.
> * Averaging across many voxels within each region/parcel, can wash out signals from a small number of activated voxels with noise from non-activated voxels.
> * Finally, Gaussian blobs provide more appealing/smoother reconstructions due to their continuous and smooth nature (a peak activation location fades smoothly from its center) when compared to averaging (which only outputs a single number for the entire parcel) for a matching number of parcels/blobs.
>
> **Interpretability of Gaussian blobs vs spatial ICA**
>
> This is true that spatial ICA tends to generate interpretable networks in task and rest-fMRI data (e.g., the well-known default-mode network in rest-fMRI) and we are not claiming otherwise.\
> However, spatial components obtained by ICA are unstructured in the sense that each ICA component may include many small and large voxel clusters across the brain (as these components are estimated unconstrained). TFA methods provide a cleaner pipeline by modeling each activation source as a Gaussian blob, therefore each TFA factor is easily interpreted through its set of parameters (i.e., its center and width parameters).\
> To make it more clear, we quote from Manning et. al. [2014b]: “Factors obtained using PCA and ICA are themselves images of the same size as the images in the original dataset (i.e., each PCA and ICA factor is a V-dimensional vector). In contrast, TFA factors are constrained to have a specified functional form... each factor is defined by a set of radial basis function parameters. Constraining TFA’s factors to have a given functional form substantially reduces the freedom TFA has to explain the dataset, which in turn reduces the fidelity of the representations. However, this reduction in reconstruction performance buys interpretability: whereas PCA and ICA factors are not directly interpretable, each TFA factor is easily interpreted through its set of parameters (e.g., its center and width parameters). In addition, TFA may be used to predict the activations of held-out voxels using their locations, whereas PCA and ICA cannot.”
>
> We will revise the corresponding sentence in the paper to clarify this.
> \
> \
> **Quantitative comparison on simulation data against HTFA, NTFA, and RKN:**
>
> Kindly note that our synthetic experiment follows that of NTFA and TFA methods and serves as a sanity check.\
> We followed up and performed a quantitative analysis against baselines for this synthetic dataset. The results showed a similar trend as in real data. We will include the results in Table 2.
>
> **Hyperparameters for real data:**
>
> In general, hyperparameters can be set according to the model’s performance in terms of held-out likelihood or prediction accuracy on a validation set.\
> Here, we chose $K=100$ in real experiments to make it roughly at par with the number of regions in AAL atlas. In practice, $K$ should be selected by a practitioner depending on the desired granularity.
> We set the size of hidden latents $z^w_t$ to be $2$ as we found it helpful for visualization purposes and because larger sizes only had a slight effect on the results.
> We selected the size of the hidden layer for the temporal transition network (i.e., $D_t$ in Table 2) from {$2,3,5$}, and picked the model with highest prediction accuracy on a small validation set ($D_t = 5$ was selected here). We set the size of the hidden layer for the temporal emission network (i.e., $D_e$ in Table 2) to the geometric mean between size of $z^w_t$ (i.e., $2$) and $K$ (size of $w_t$), which is $D_e = \sqrt{2\times 100} \sim 15$.
> \
> \
> **Model for estimating likelihood? Is likelihood estimated per subject?**
>
> The likelihood is intractable. However, we compute an importance sampling based lower-bound approximation of the likelihood which is also adopted in NTFA (Sennesh et. al., NeurIPS'21, see their section 4.3 and equation 11), DMM (Krishnan et.al., AAAI’17, https://arxiv.org/pdf/1609.09869.pdf, see equations 9, 10 in their Appendix A), and Auto-encoding variational bayes (Kingma and Welling, ICLR’14, Appendix D).
>
> In brief, assuming $\tilde{Y}$ to be a test data and $Y$ to be the train data, we approximate posterior-predictive likelihood from $L=100$ samples (superscript $(l)$ denotes a sample):
>
> $$p(\tilde{Y}|Y) \geq E[\tilde{\mathcal{L}}] =\frac{1}{L}\sum_l \sum_t \log p_\theta(\tilde{Y}_t | \tilde{w}_t^{(l)}, \tilde{\rho}^{(l)}, \tilde{\gamma}^{(l)}, \tilde{z}_t^{w(l)}, \tilde{z} _{t-1}^{w(l)}, z^{F(l)}, C^{(l)}),$$
>
> We sample $z^w_{t-1}$, $z^F$, and $C$ from their variational distribution and the remaining latent variables from their prior (i.e., generative model):
>
> * $\tilde{z}^{w(l)}_{t-1}\sim q(\tilde{z}^w _{t-1}),\quad z^{F(l)} \sim q(z^F),\quad C^{(l)} \sim q(C)$
>
> * $\tilde{\rho}^{(l)}, \tilde{\gamma}^{(l)} \sim p(\tilde{\rho}, \tilde{\gamma}|z^{F(l)}), \quad \tilde{z}_t^{w(l)} \sim p(\tilde{z}_t^w | \tilde{z}^{w(l)} _{t-1}), \quad \tilde{w}_t^{(l)} \sim p(\tilde{w}_t|\tilde{z}_t^{w(l)}).$
>
> The likelihood is estimated per subject and the average is reported in Table 2. We will add their standard deviations to this table. DMFA consistently performs better across all subjects in terms of predictive log-likelihood, so the difference is significant in a statistical sense.
> \
> \
> **Would ST-GCN fall under the scope of this paper?**
>
> The ST-GCN (e.g., paper of https://arxiv.org/pdf/2003.10613.pdf) works on ROI fMRI data (they used 22 and 34 brain regions in their paper); so in this sense, it is different from DMFA and TFA methods which directly process high-dimensional raw fMRI data (therefore, this method does not lie in the category of factorization).\
> Also, in contrast to DMFA, ST-GCN does not learn a temporal transition model and therefore cannot be used for temporal prediction. We will add a description of this method to the paper.
>
> **More description on TFA**
>
> We will include a detailed description of TFA methods in the supplementary.
>
> **Multiple comparison correction**
>
> Thanks for pointing this out.\
> DMFA performed better on Caltech (p=0.0033), SBL (p=1.3e-6), Stanford (p=9.0e-6), Yale (p=9.5e-4), and Full dataset (p=3.1e-4) (MaxMun did not pass p<0.005 threshold), and ROI/ICA+SVM performed better on NYU (p=1.2e-6).

---

### Official Review · Reviewer_FHJp · 2021-07-19

**Rating:** 6
**Confidence:** 4

**Summary:**

This paper introduces Deep Markov Factor Analysis (DMFA). DMFA can model spatiotemporal BOLD fMRI time series by using time-varying latent factors that govern time-varying spatial maps consisting of spatial mixtures of RBFs. Latent time series are associated to a categorical clustering variable in order to enable grouping. Certain links in the Bayesian diagram are modeled using deep neural networks. Optimization is done by a variational approach.
Experiments are performed on a simulated data set with known ground truth, the ABIDE autism data set, and a Depression study data set. Some clustering of relevant clinical outcomes is observed.




**Limitations And Societal Impact:**

There is no reason to believe there would be a negative societal impact to this work. That there would be a strongly positive one is also unlikely.

Limitations were not really discussed.

**Main Review:**

DMFA lies at the intersection of the Topographical Factor Analysis approach (ref 14), unsupervised TV-regularized resting-state analysis as performed in (https://www.nature.com/articles/ncomms8751) and variational autoencoders for time series (https://arxiv.org/abs/1506.02216). The latter two papers are not mentioned.

Several concerns arise around this work:

What is the point in using Gaussian blobs to model spatial maps? The argument that the idea is inherited from TFA is insufficient. Why not just use voxel maps at a certain resolution? Not all brain regions are shaped like spherical Gaussians or sparse sums thereof. The reconstructed maps show this very clearly -- pretty much all detail is lost.

What is the point of *temporally varying* spatial Gaussian blobs? Surely brain regions remain in place over time. One could make the argument that “activity moves around”, but it remains true that certain spatially fixed regions receive more or less blood over time. It seems that this feature adds unnecessary complexity and hinders interpretability

The synthetic model is not very helpful since it basically generates data according to the model used. Why not generate some fake brain data based on a different model, like ICA components?

The experimental evaluation does not seem to contain any clearly stated numbers enabling comparisons to other methods. It is great that the method finds some factors in an unsupervised manner, but please state the accuracy over the full datasets for the task. It may be helpful to look at Abraham et al 2017 (ref from the paper) for the data curation procedure as this is an important thing to get right.

Why is there no comparison to the method presented in (https://www.nature.com/articles/ncomms8751), adapted to this task? This provides a simpler baseline.


**Time Spent Reviewing:**

3

---

> ### Author Response · Authors · 2021-08-10
> **Thank you.**
>
> Thanks for your valuable feedback.
>
> Thanks for introducing references of https://www.nature.com/articles/ncomms8751 and https://arxiv.org/abs/1506.02216. The latter lies in the category of amortized variational inference for sequential data analysis. We will cite them and add their discussion to the paper.
>
> **Why Gaussian blobs?**
>
> * The smoothness (or loss of details) in images is not specific to DMFA. Averaging the signals over voxels maps will also result in loss of details (as you are assigning a single value to each parcellation). More detailed reconstructions can be obtained by choosing a larger number of Gaussian blobs (or parcellations). In fact, Gaussian blobs provide more appealing estimations due to their continuous and smooth nature when compared to averaging (which only outputs a single number) for a matching number of parcels.
> * We agree that voxel maps can be very helpful, but these maps are often obtained from external datasets and are assumed to be global across all subjects, so they smooth out intra-subject variability. By using Gaussian blobs of the same or more resolution (and by automatically learning their locations and extents) we let the model determine significant activation locations and also preserve subject-level variability.
> * This is true that brain regions are not necessarily spherical Gaussians, but an arbitrary shaped brain region can be roughly captured by superimposition of enough Gaussian blobs. Also, there could be a case that a brain region includes multiple activation sources. This can be captured by multiple Gaussian blobs, but could be lost by averaging across the entire region (which is usually done in voxel maps). In addition, averaging across many voxels within each region, can wash out signals from a small number of activated voxels with noise from non-activated voxels.
>
> **Temporally varying Gaussian blobs?**
>
> We believe that there is a slight confusion here. The spatial Gaussian blobs are not varying across time (they are fixed over time and are not indexed by time), but their mixing weight is changing over time, similar to previous methods. However, we allow Gaussian blobs to slightly perturb across different data points (i.e., subjects, which are indexed by n). This slight perturbation is controlled by a prior over location and extent of Gaussian blobs: $p(\rho_n, \gamma_n | z^F_n)$, where $z^F_n \sim \text{Norm}(0,\text{I})$.
> \
> \
> **ICA components for synthetic model:**
>
> The synthetic experiment follows that of NTFA and TFA methods and serves as a sanity check.
>
> We followed up and generated synthetic data based on 30 spatial ICA components obtained from the Autism Caltech dataset. The rest of the experiment is the same as our synthetic experiment in the paper, except for the fact that we used K=100 spatial components to fit on the model. Again, we observed the same clustering effect in our temporal latent. This is expected because ICA components can be approximated with sum of multiple Gaussian blobs, and from there the rest of the pipeline should work as before.
>
> **Accuracy over the full Autism dataset:**
>
> The accuracy for DMFA+SVM over the full Autism dataset is 64%$\pm$3. We will add a table containing the accuracy numbers for all subsets of this dataset and all methods in the supplementary.
>
> **Comparison with https://www.nature.com/articles/ncomms8751:**
>
> Thank you for introducing this baseline. We couldn’t find an open-source implementation of this paper. We will implement their method and add comparison results to the final version of our paper.

---

> > ### Comment · Reviewer_FHJp · 2021-08-28
> >
> > Thanks for the clarifications. I seem to have misinterpreted some indices leading to the assumption of time-variability of the blobs.
> >
> > Regarding subject-specific maps, this is great, but also totally achievable with voxel maps. See the cited Abraham 2017 for a two-level multi-subject dictionary-learning procedure, which also tends to learn local blobs.
> >
> > I raise my score to 6.

---

### Official Review · Reviewer_mRP4 · 2021-07-21

**Rating:** 7
**Confidence:** 4

**Summary:**

The authors present deep Markov factor analysis (DMFA) to capture temporal dynamics in functional magnetic resonance imaging data. The method chosen by the authors is relevant since existing methods overlook the highly nonlinear and complex temporal dynamics of neural processes when factorizing their imaging data. The authors claim that their method (DMFA) is able to cluster fMRI data in its low dimensional temporal embedding, which enables validation of fMRI related hypotheses.

**Limitations And Societal Impact:**

Summary:

The authors explored a relevant problem and can be widely used in neuroimaging to transfer high dimensional imaging data into low dimensional and interpretable representations.

**Main Review:**

Reasons to accept:

- If the authors claims are correct, the study is ground breaking in fMRI data analysis since the current models do not encode temporal dynamics. The voxel activations have non-linearity and inherent temporal dependencies, which are explored using the methods mentioned by authors.

- The paper is well structured and explained. The theory behind Deep Markov Factor Analysis (DMFA) is explained perfectly.

- The authors thoroughly explored the current state-of-the art related to the paper. The comparisons are clear in the text, table as well as the figures.

- Limitations an ablation study are explained clearly.

- Ablation study is clearly explained by the authors.



Reasons to decline:

- Are the claims mentioned in following lines correct? The authors mentioned in line number 95-98 that RNN-based methods do not estimate any spatial factors, and are harder to

train on long sequences. Is this verified on Convolutional RNN models (Wang et al., 2019) and LSTM/GRU models (Yan et al., 2019)? Are there no recent papers using transformer based models?

- Better visualizations. Hard to interpret Figure 2 (b) and Figure 3 (Middle) images.

**Time Spent Reviewing:**

72

---

> ### Author Response · Authors · 2021-08-10
> **Thank you.**
>
> Thanks for your valuable feedback.
>
> Thanks for highlighting the novelty, significance, and clarity of our work, which is very encouraging.
>
> **RNN-based methods:**
> * Although Convolutional RNN models capture both temporal and spatial correlations, by construction, they do not provide the familiar spatial correlation maps that we expect in neuroimaging analysis and that are necessary for neuroscientific interpretability.
> * While RNN models suffer from the vanishing/exploding gradients problem (and are harder to train on long sequences), this problem is much alleviated in LSTM/GRU/Transformer architectures. However, LSTM/GRU architectures still result in a huge computational graph and intense GPU memory consumption on long sequences, which lead to very long training times. \
> DMFA samples from a distinct posterior distribution at each time point and feeds that to a neural network for estimating distribution parameters of the prior for the next time point. This framework accommodates arbitrary length sequences and allows parallelization of computations over time for a faster training process.
>
> We will revise the sentence in lines 95-98 to clarify this.
>
> **Better visualizations:**
>
> We will add color bars to Figure 2 (b) and Figure 3 (Middle) for a better readability.

---

### Official Review · Reviewer_tynp · 2021-07-21

**Rating:** 7
**Confidence:** 3

**Summary:**

The paper presents a deep Markov factor analysis (DMFA), which is a generative model that employs Markov property in a chain of low dimensional temporal embeddings as well as spatial inductive assumptions in order to capture temporal dynamics in functional magnetic resonance imaging (fMRI). The paper shows that DMFA has capability to cluster fMRI data in its low temporal embedding in regards to subject and cognitive state variability. The advantages of DMFA are demonstrated both through synthetic and application data.

**Limitations And Societal Impact:**

Yes

**Main Review:**

The paper meets and exceeds the criteria of originality, quality, clarity and significance. The paper clearly indicates previous methodologies in the direction of their method (i.e., factor analysis and dynamic factorization), while providing ample citations for each. The development of DMFA is clearly presented with explicit and easy-to-understand diagrams. The estimation method (variational inference) is also clearly presented and makes it easy to reproduce for any interested practitioner. The simulation experiments show the clear utility of the  method for the generative model to learn from complex spatiotemporal dynamics presented in fMRI data. And finally, several application datasets are presented (e.g., in Autism, Depression, Ablation, etc.). The reviewer sees no major issues with the current work, and sees the paper as well-polish and an excellent example of a high-quality submission.

But due to some comments from other reviewers, I do see some significant issues with methodology/innovation, as well as a lack of proper sensitivity analysis. Therefore I will reduce my score.

**Time Spent Reviewing:**

0.5

---

> ### Author Response · Authors · 2021-08-10
> **Thank you.**
>
> Thanks for your encouraging feedback.
>
> We are happy to see that the reviewer valued the novelty, significance, clarity, and reproducibility of our work.
>
> Thanks again for highlighting those points.

---

> ### Author Response · Authors · 2021-08-27
> **Follow-up**
>
> Thanks again for your feedback.
>
> We kindly ask the reviewer to check our follow-up discussion with "Reviewer 1fHu" here: https://openreview.net/forum?id=ekVPXh9tYkL&noteId=LbSuc2qzc2L regarding our methodology (to see if any of their concerns is addressed)
>
> We will be happy to address any specific concerns the reviewer may have regarding our methodology.
>
> Regarding the sensitivity analysis, we will add sensitivity curves for $D_t$ (the size of the hidden layer for the temporal transition network) and $\sigma_0$ (observation noise) to the supplementary. Please note that DMFA shows little sensitivity to these parameters.

---

> > ### Comment · Reviewer_tynp · 2021-09-10
> > **reviewer reply**
> >
> > Great! Thank you for addressing my concerns. I will modify my review accordingly.

---

### Decision · Program_Chairs · 2021-09-27

**Decision:**

Accept (Poster)

**Comment:**

Dear authors,

the overall consensus is that your work has merit in terms of method and experiments
although there are a number of concerns about some limited comparison with alternative
baseline approaches. Despite this concern I am inclined to flip the decision on the
positive side as code is available and you can still insert some numbers in the camera
ready as proposed.

Best regards,
The AC